



# Chemical composition of nanoparticles from α-pinene nucleation and the influence of isoprene and relative humidity at low temperature

Lucía Caudillo[1], Birte Rörup[2], Martin Heinritzi[1], Guillaume Marie[1], Mario Simon[1], Andrea C. Wagner[3], Tatjana Müller[1,4], Manuel Granzin[1], Antonio Amorim[5], Farnoush Ataei[6], Rima Baalbaki[2], Barbara Bertozzi[7], Zoé Brasseur[2], Randall Chiu[3], Biwu Chu[2], Lubna Dada[8], Jonathan Duplissy[2,9], Henning Finkenzeller[3], Loïc Gonzalez Carracedo[10], Xu-Cheng He[2], Victoria Hofbauer[11], Weimeng Kong[12,13], Houssni Lamkaddam[8], Chuan P. Lee[8], Brandon Lopez[11], Naser G. A. Mahfouz[11], Vladimir Makhmutov[14,26], Hanna E. Manninen[15], Ruby Marten[8], Dario Massabò[16], Roy L. Mauldin[17,11], Bernhard Mentler[18], Ugo Molteni[8,20,21], Antti Onnela[15], Joschka Pfeifer[15], Maxim Philippov[14], Ana A. Piedehierro[22], Meredith Schervish[11], Wiebke Scholz[18], Benjamin Schulze[12], Jiali Shen[2], Dominik Stolzenburg[2], Yuri Stozhkov[14], Mihnea Surdu[8], Christian Tauber[10], Yee Jun Tham[2], Ping Tian[23], António Tomé[24], Steffen Vogt[7], Mingyi Wang[11], Dongyu S. Wang[8], Stefan K. Weber[15], André Welti[22], Wang Yonghong[2], Wu Yusheng[2], Marcel Zauner-Wieczorek[1], Urs Baltensperger[8], Imad El Haddad[8], Richard C. Flagan[12], Armin Hansel[18,19], Kristina Höhler[7], Jasper Kirkby[1,15], Markku Kulmala[2,9,25], Katrianne Lehtipalo[2,22], Ottmar Möhler[7], Harald Saathoff[7], Rainer Volkamer[3], Paul M. Winkler[10], Neil M. Donahue[11], Andreas Kürten[1], Joachim Curtius[1].

[1]Institute for Atmospheric and Environmental Sciences, Goethe University Frankfurt, 60438 Frankfurt am Main, Germany
[2]Institute for Atmospheric and Earth System Research (INAR) / Physics, Faculty of Science, University of Helsinki, 00014 Helsinki, Finland
[3]Department of Chemistry & CIRES, University of Colorado Boulder, Boulder, CO, 80309-0215, USA
[4]Max Planck Institute for Chemistry, Mainz, 55128, Germany
[5]CENTRA and FCUL, University of Lisbon, 1749-016 Lisbon, Portugal
[6]Leibniz Institute for Tropospheric Research, Leipzig, 04318, Germany
[7]Institute of Meteorology and Climate Research, Karlsruhe Institute of Technology, 76344 Eggenstein-Leopoldshafen, Germany
[8]Laboratory of Atmospheric Chemistry, Paul Scherrer Institute, 5232 Villigen, Switzerland
[9]Helsinki Institute of Physics, University of Helsinki, 00014 Helsinki, Finland
[10]Faculty of Physics, University of Vienna, 1090 Vienna, Austria
[11]Center for Atmospheric Particle Studies, Carnegie Mellon University, Pittsburgh, PA, 15213, USA
[12]Division of Chemistry and Chemical Engineering, California Institute of Technology, Pasadena, CA 91125, USA
[13]California Air Resources Board, Sacramento, CA 95814, USA
[14]Lebedev Physical Institute, Russian Academy of Sciences, 119991, Moscow, Russia
[15]CERN, 1211 Geneva, Switzerland
[16]Dipartimento di Fisica, Università di Genova and INFN, 16146 Genova, Italy
[17]Department of Atmospheric and Oceanic Sciences, University of Colorado Boulder, Boulder, CO 80309, USA
[18]Institute for Ion and Applied Physics, University of Innsbruck, 6020 Innsbruck, Austria
[19]Ionicon Analytik GmbH, 6020 Innsbruck, Austria
[20]Forest Dynamics, Swiss Federal Institute for Forest, Snow and Landscape Research, 8903 Birmensdorf, Switzerland
[21]Department of Chemistry, University of California, Irvine, Irvine, CA 92697, USA
[22]Finnish Meteorological Institute, 00560 Helsinki, Finland
[23]Beijing Weather Modification Office, China
[24]IDL, Universidade da Beira Interior, R. Marquês de Ávila e Bolama, Covilhã, 6201-001, Portugal





[25]Aerosol and Haze Laboratory, Beijing Advanced Innovation Center for Soft Matter Science and Engineering, Beijing
University of Chemical Technology, Beijing, 100029, P.R. China
[26]Moscow Institute of Physics and Technology (National Research University), 117303, Moscow, Russia

*Correspondence to*: Lucía Caudillo (lucia.caudillo@iau.uni-frankfurt.de) and Joachim Curtius (curtius@iau.uni-frankfurt.de)

**Abstract.** New Particle Formation (NPF) from biogenic organic precursors is an important atmospheric process. One of the major species is α-pinene, which upon oxidation, can form a suite of products covering a wide range of volatilities. A fraction of the oxidation products is termed Highly Oxygenated Organic Molecules (HOM). These play a crucial role for nucleation and the formation of Secondary Organic Aerosol (SOA). However, measuring the composition of newly formed particles is challenging due to their very small mass. Here, we present results on the gas and particle phase chemical composition for a system where α-pinene was oxidized by ozone, and for a mixed system of α-pinene and isoprene, respectively. The measurements took place at the CERN Cosmics Leaving Outdoor Droplets (CLOUD) chamber at temperatures between -50 °C and -30 °C and at low and high relative humidity (20 % and 60 to 100 % RH). These conditions were chosen to simulate pure biogenic new particle formation in the upper free troposphere. The particle chemical composition was analyzed by the Thermal Desorption-Differential Mobility Analyzer (TD-DMA) coupled to a nitrate chemical ionization time-of-flight mass spectrometer. This instrument can be used for particle and gas phase measurements using the same ionization and detection scheme. Our measurements revealed the presence of $C_{8-10}$ monomers and $C_{18-20}$ dimers as the major compounds in the particles (diameter up to ~ 100 nm). Particularly, for the system with isoprene added, $C_5$ ($C_5H_{10}O_{5-7}$) and $C_{15}$ compounds ($C_{15}H_{24}O_{5-10}$) are detected. This observation is consistent with the previously observed formation of such compounds in the gas phase. However, although the $C_5$ and $C_{15}$ compounds do not easily nucleate, our measurements indicate that they can still contribute to the particle growth at free tropospheric conditions. For the experiments reported here, most likely isoprene might enhance growth at particle sizes larger than 15 nm. Besides the chemical information regarding the HOM formation for the α-pinene (plus isoprene) system, we report on the nucleation rates measured at 1.7 nm and found that the lower $J_{1.7nm}$ values compared with previous studies are very likely due to the higher α-pinene and ozone mixing ratios used in the present study.

# 1 Introduction

Approximately half of the global Cloud Condensation Nuclei (CCN) are produced by nucleation (Merikanto et al., 2009; Gordon et al., 2017). In particular, biogenic emissions of Volatile Organic Compounds (VOCs) play an important role in the formation of aerosol particles. The chemical reactions involving VOCs can lead to the formation of Highly Oxygenated Organic Molecules (HOM), which can be described as a class of organic compounds that are formed under atmospherically relevant conditions by gas phase autoxidation involving peroxy radicals (Ehn et al., 2014; Bianchi et al., 2019). These compounds possess low saturation vapor pressures and are thus relevant for New Particle Formation (NPF) and Secondary Organic Aerosol (SOA) formation due to gas-to-particle conversion.



One of the most prominent biogenic precursors for the formation of particulate material is α-pinene ($C_{10}H_{16}$). It is known that α-pinene oxidation forms HOM that have the ability to nucleate on their own under atmospheric conditions, without the involvement of other trace gases, e. g., sulfuric acid (Kirkby et al., 2016; Tröstl et al., 2016). Stolzenburg et al. (2018) showed that the rapid growth of organic particles produced by α-pinene dark ozonolysis at +25 °C, +5 °C, and -25 °C is determined by the lower extent of autoxidation at reduced temperatures and the decrease in volatility of all oxidized molecules.

Furthermore, Simon et al. (2020), extended the study of α-pinene gaseous oxidation products to even lower temperatures from +25 °C to -50 °C, showing that the oxygen to carbon ratio (O:C) and the yield for HOM formation decrease as the temperature decreases, whereas the reduction of volatility compensates this effect by increasing the nucleation rates at lower temperatures.

Isoprene ($C_5H_8$) is the biogenic vapor with the highest global emission rate. Its estimated emissions are between 500 to 600 Tg per year (Guenther et al., 2006; Sindelarova et al., 2014) and there are many studies that indicate the global

importance of isoprene in terms of SOA formation (Surratt et al., 2006; Surratt et al., 2007; Surratt et al., 2010; Paulot et al., 2009; Lin et al., 2012; Riva et al., 2016). Kiendler-Scharr et al. (2009) presented observations at 15 ℃ of a significant decrease in particle number and volume concentration by the presence of isoprene in an experiment under plant-emitted VOCs conditions. Subsequently, McFiggans et al. (2019) showed that isoprene, carbon monoxide, and methane can each suppress aerosol mass and the yield from monoterpenes in mixtures of atmospheric vapors.

Recently, a study by Heinritzi et al. (2020) revealed that the presence of isoprene in the α-pinene system suppresses new particle formation by altering the peroxy-radical termination reactions and inhibiting the formation of those molecules needed for the first steps of cluster and particle formation (species with 19 to 20 carbon atoms). For these biogenic systems, α-pinene and α-pinene + isoprene, the mechanisms behind the formation of HOM in the gas phase have been studied over a wide temperature range. However, the particle phase has not been characterized to the same extent because of the difficulties

in measuring the nanoparticle chemical composition due to their very small mass. Despite that, there have been several efforts for designing and improving techniques to face this problem.

Some particle phase studies exist that report the chemical composition of newly formed nanoparticles. For instance, Kristensen et al. (2017), measuring at 293 K and 258 K, showed an increased contribution of less oxygenated species to α-pinene SOA particles formed from ozonolysis at sub-zero temperatures. Ye et al. (2019) measured the particle phase chemical

composition from α-pinene oxidation between -50 °C and +25 °C with the FIGAERO (Lopez-Hilfiker et al., 2014). They found that during new particle formation from α-pinene oxidation, gas phase chemistry directly determines the composition of the condensed phase. Highly Oxygenated Organic Molecules are much more abundant in particles formed at higher temperatures, shifting the compounds towards higher O:C and lower volatilities. Additionally, some studies addressing the chemical composition, volatility, and viscosity of organic molecules have provided important insights into their influence on

the climate (Huang et al., 2018; Reid et al., 2018; Champion et al., 2019).

Here, we present the results from gas and particle phase chemical composition measurements for a system where α-pinene was oxidized to simulate pure biogenic new particle formation at free tropospheric conditions in a range from -50 °C to -30 °C. The data are further compared to the mixed system of α-pinene and isoprene in order to better understand the





partitioning processes. The particle chemical composition was analyzed by the Thermal Desorption-Differential Mobility

Analyzer (TD-DMA) (Wagner et al., 2018), coupled to a nitrate chemical ionization time-of-flight mass spectrometer. This technique allows a direct comparison between gas and particle phase as both measurements are using the identical chemical ionization source and detector.

## 2. Methods

### 2.1 The CLOUD chamber at CERN and the experiments

The measurements took place in the Cosmics Leaving Outdoor Droplets (CLOUD) chamber at the European Organization for Nuclear Research (CERN) during the CLOUD14 campaign (Sep - Nov 2019). The CLOUD chamber is a stainless-steel cylinder, with a volume of 26.1 m³, which has been built to the highest technical standards of cleanliness (Kirkby et al., 2011; Duplissy et al., 2016). By precisely controlling several parameters such as, gas concentrations, temperature, relative humidity, ultraviolet light intensity and internal mixing, specific atmospheric systems can be recreated in order to study the nucleation

and growth processes of aerosols at atmospheric conditions. The biogenic gas concentrations, here α-pinene and isoprene, can be regulated by using individual evaporator supplies, in which dry nitrogen passes through the evaporator containing the precursors in a liquid form, at controlled temperature. In this way, the precursors are evaporated and diluted with clean air to achieve the desired concentration in the chamber. Ozone is introduced via a separate gas line. The chamber is continuously stirred by two magnetically coupled stainless-steel fans placed at the top and at the bottom of the chamber to provide a

homogeneously mixed system (Voigtländer et al., 2012).

The experiments relevant for this work were done in a through mode with continuous addition of the reactants and performed at -50 °C and -30 °C and at low and high relative humidity to simulate pure biogenic new particle formation at a range of free tropospheric conditions. Isoprene and α-pinene precursor gases were oxidized with $O_3$ and ˙OH (produced from $O_3$ photolysis in the presence of $H_2O$ and UV light) to induce both dark ozonolysis and photochemistry oxidation reactions.

The α-pinene level was between 1 and 8 ppbv, the isoprene level up to 30 ppbv, and $O_3$ approximately 100 ppbv. The ozonolysis of α-pinene was performed at -50 °C and -30 °C, while the α-pinene + isoprene experiment was performed at -30 °C only. The experimental overview is discussed in more detail in Sect. 3.1.

### 2.2 TD-DMA

The particle chemical composition was analyzed by the Thermal Desorption-Differential Mobility Analyzer (TD-DMA)

coupled to a nitrate chemical ionization time-of-flight mass spectrometer. The TD-DMA design and characterization have been described in detail by Wagner et al. (2018). This instrument allows the direct comparison between gas and particle phase chemical composition as both measurements use the same ionization scheme and mass spectrometer (the detection technique will be described in Sect. 2.3).



The TD-DMA uses an online and semi-continuous principle for the detection of the chemical composition of nanoparticles. The particles are sampled from the chamber, charged with an X-ray source, a specific size can be selected and immediately afterwards they are electrostatically collected on a filament. Heating the filament after a defined collection time evaporates the particles into a stream of clean carrier gas ($N_2$). The particle vapor is analyzed by the nitrate CI-APi-TOF mass spectrometer (Kürten et al., 2014). In order to estimate the instrumental background, two heating profiles are recorded: the first heating cycle evaporates all the particulate material collected; a second heating cycle constrains the background due to the heating of the inlet line. All reported particle phase signals are corrected based on this background measurement.

For the experiments that are reported in this work, a filament of platinum/rhodium (90:10) was used, and an integral, nonsize - selective mode of operation was chosen in order to maximize the mass of collected particles. Due to the very low experimental temperatures, cold sheath flows and isolated inlet lines were installed in order to avoid drastic temperature changes between the CLOUD chamber and the instrument. Evaporation of particulate material before the analysis should therefore not be substantial.

### 2.3 Nitrate CI-APi-TOF mass spectrometer

The gas phase and the evaporated particulate material were measured using a nitrate chemical ionization atmospheric-pressure-interface time-of-flight (CI-APi-TOF) mass spectrometer, which has three major components: an atmospheric pressure ion-molecule reactor where the chemical ionization takes place; an atmospheric pressure interface for transporting the charged ions into the mass classifier; and a time-of-flight mass classifier where the ions are accelerated, separated according to their mass-to-charge ratio and detected with a microchannel plate (Jokinen et al., 2012; Kürten et al., 2014). The nitrate CI-APi-TOF mass spectrometer uses nitrate reagent ions $(HNO_3)_n NO_3^-$ with $n = 0-2$, which are created by an ion source using a corona discharge needle (Kürten et al., 2011). With this nitrate chemical ionization technique, sulfuric acid, iodic acid, dimethylamine and HOM can be detected (Kürten et al., 2014; Simon et al., 2016; Kirkby et al., 2016; He et al., 2021). HOM are detected because of the presence of functional groups such as hydroperoxy (-OOH) or hydroxy (-OH), which provide the hydrogen bonds required for clustering with the reagent ions.

Here the nitrate CI-APi-TOF mass spectrometer data for gas and particle phase have been corrected for background signals and the mass-dependent transmission efficiency in the mass classifier (Heinritzi et al., 2016). The data analysis and processing were performed using IGOR Pro 7 (WaveMetrics, Inc., USA), Tofware (Version 3.2, Aerodyne Inc., USA) and MATLAB R2019b (MathWorks, Inc., USA).

### 2.4 Formation rates

The particle number size distribution between ~ 1 nm and 1 μm is measured using a suite of particle counters namely a particle size magnifier, PSM (Vanhanen et al., 2011), a condensational particle counter (CPC 3776, TSI), a nano scanning mobility particle sizer (nano-SMPS 3982, TSI), and a home-built long scanning mobility particle sizer (long-SMPS). The PSM measures the size distribution between ~1 and 3 nm as well as the total particle number concentration above a defined cutoff, 1.7 nm in



this study. The CPC on the other hand is used to measure the total particle number concentration above 2.5 nm. The nano-SMPS and long-SMPS together cover the particle number size distribution between 6 nm and 1 μm. The same set-up has been used in previous CLOUD experiments, see for example Lehtipalo et al. (2018) and Heinritzi et al. (2020).

The particle formation rate ($J_{dp}$), which is defined as the flux of particles of a certain size as a function of time, is

calculated using the method proposed by Dada et al. (2020), see equation (9) therein. For this study, the formation of particles with a diameter ≥ 1.7 nm is calculated ($J_{1.7}$) using the derivative of the total concentration of particles measured with the PSM while accounting for size-dependent losses to the chamber wall, by coagulation or via dilution. The error on $J_{1.7}$ is 30% based on run-to-run repeatability (Dada et al., 2020).

## 3. Results and discussion

**3.1 Experimental overview**

An overview of the experiments performed at -30 °C and -50 °C at low and high relative humidity is shown in **Fig. 1**. The mixing ratio of ozone was stable at ~ 100 ppbv for all of the experiments reported in this work (not shown). In order to represent pure biogenic new particle formation events, no other trace gases were added to the chamber and the levels of $SO_2$, $NO_x$, and other trace gases were monitored to remain always below the detection limits of the respective measurement devices. By using

the TD-DMA, particles were collected in every NPF system (without resolving the particle size), the shaded area in **Fig. 1** refers to the period where the particle collection took place.

The upper panel of **Fig. 1** displays the size distribution measured by the Scanning Mobility Particle Sizer (SMPS). Four different experiments can be categorized as follows:

1. α-pinene + isoprene at -30 °C and 20 % RH (*αIP-30,20*);

2. α-pinene at -30 °C and 20 % RH (*α-30,20*);

     3. α-pinene at -50 °C and 20 % RH (*α-50,20*); and,

     4. α-pinene at -50 °C and 60 to 100 % RH (*α-50,60-100*).

The color scale in the upper panel of **Fig. 1** indicates that the newly-formed particles appear in the smallest size channels of the SMPS soon after the concentration of α-pinene in the chamber is increased. The experiments were performed such that the

particles grew rapidly to reach sizes of approximately 100 nm, where they could potentially act as Cloud Condensation Nuclei (CCN) or Ice Nucleating Particle (INP) and were used for further CCN and INP studies.

The second panel of **Fig. 1** shows the particle number concentration measured by the Condensation Particle Counter ($CPC_{2.5nm}$) and by the Particle Size Magnifier (PSM) with a cut-off diameter of 1.7 nm. A higher particle number concentration can be observed for the experiments *α-50,20* and *α-50,60-100* at -50 °C, reaching ~ 2 x $10^5$ $cm^{-3}$. By comparing the experiments

at -30 °C, *α-30,20* and *αIP-30,20* a lower particle concentration is observed for the system where isoprene is present. The reduction is approximately a factor of 3; this can be attributed to the suppression of the new particle formation by isoprene oxidation. This confirms the results of Kiendler-Scharr et al. (2009), who first reported the decrease in particle number of the





nucleated particles. The effect of isoprene in terms of total HOM concentration in the gas phase and on the measured new particle formation rates will be discussed in more detail in Sect. 3.4.2.

205        The third panel of **Fig. 1** shows the α-pinene and isoprene mixing ratios. For all of the systems, α-pinene was between 1 and 8 ppbv, while isoprene was only present during experiment *αIP-30,20* up to 30 ppbv. The precursor gases were measured by using a proton transfer reaction time-of-flight (PTR-TOF) mass spectrometer (Graus et al., 2010; Breitenlechner et al., 2017), which is capable of measuring VOCs.

        The bottom panel of **Fig. 1** shows the total HOM concentration in the gas phase. Here, the total HOM is defined as

the sum of $C_5$, $C_{10}$, $C_{15}$ and $C_{20}$ carbon classes; these classes consider compounds with $C_2$ - $C_5$, $C_6$ - $C_{10}$, $C_{11}$ - $C_{15}$ and $C_{16}$ - $C_{20}$, respectively and considered as a HOM such compounds with five or more oxygen atoms as suggested in Bianchi et al. (2019). The total HOM was measured with a calibrated nitrate CI-APi-TOF mass spectrometer (Kürten et al., 2012). Additionally, a temperature dependent sampling loss correction factor is applied. From the evolution of these traces, it can be observed that, $C_5$ and $C_{15}$ carbon classes have higher concentrations (approximately by a factor of 2.5) in experiment *αIP-30,20* compared

with *α-30,20*, which can be explained by the presence of isoprene. However, possible fragmentation in the α-pinene ozonolysis systems also can lead to some $C_5$ and $C_{15}$ compounds produced without the presence of isoprene.

## 3.2 Gas and particle phase chemical composition

**Figure 2** shows the carbon distribution as an overview of the compounds detected in gas and particle phase for a system where only α-pinene was oxidized (*α-30,20*). $C_{8-10}$ monomers (**Fig. 2a**) and $C_{18-20}$ (**Fig. 2b**) dimers are observed in the gas as well as

in the particle phase. For instance, some of the signals with the highest intensity correspond to $C_{10}H_{16}O_{3-9}$, and $C_{20}H_{32}O_{5-13}$, especially $C_{10}H_{16}O_6$ and $C_{10}H_{16}O_7$ have an important presence in both phases. Overall, most of the compounds that are present in the gas phase are detected as well in the particle phase, although their relative contribution to the total signal can differ between the phases.

### 3.2.1 Influence of isoprene on α-pinene system at -30 °C and 20 % RH

**Figure 3** shows mass defect plots of gas and particle phase and the intensity difference between them for the experiments at -30 °C. **Figure 3a** and **Fig. 3d** display the gas and particle of α-pinene at -30 °C and 20 % RH (*α-30,20*). While the gas and particle of α-pinene + isoprene at -30 °C and 20 % RH (*αIP-30,20*) are shown in **Fig. 3b** and **Fig. 3e**, respectively. As both phases were measured with the same instrument, they can be directly inter-compared.

        The intensity difference is calculated based on the normalized signal (each single signal divided by the total signal

for each system and phase). Essentially, the normalized signal can be understood as a measure of the fraction or contribution of every compound in the entire system. By looking at the intensity difference in the gas phase (**Fig. 3c**), it can be observed that some $C_5$ and $C_{15}$ contribute significantly more in the system with isoprene added (*αIP-30,20*) that are not as pronounced in the system where only α-pinene was oxidized (*α-30,20*). This observation can be attributed to the presence of isoprene in the system. As described by Heinritzi et al. (2020), $C_{15}$ dimers are formed in the gas phase when $C_{10}$ $RO_2$· radicals from α-

https://doi.org/10.5194/acp-2021-512


pinene ozonolysis undergo terminating reactions with $C_5 RO_2\cdot$ radicals from the isoprene oxidation with $\cdot OH$. Additionally, $C_{19}$ and $C_{20}$ dimers contribute more in the system where only α-pinene was oxidized (*α-30,20*).

      **Fig. 3f** shows the intensity difference in the particle phase. This indicates that there is an enhancement of $C_{4-5}$ and $C_{13-16}$ compounds in the system with isoprene (*αIP-30,20*). There is especially a group of $C_{15}$ compounds $C_{15}H_{24}O_{5-10}$ (see **Fig. S1** in the supplement) with significant signal in the particle phase. Since the particles collected reached sizes up to ~ 100 nm,

and it has been shown in previous studies (Heinritzi et al., 2020) that $C_{15}$ dimers do contribute to the growth, this observation confirms the existence of these species in the condensed material. Certainly, isoprene can suppress new particle formation (it will be discussed in Section 3.4.2). However, isoprene can still contribute to the growth of particles by $C_5$ or by $C_{15}$ compounds. Thus, these species can be an important fingerprint to identify SOA from a mixture of biogenic vapors containing isoprene.

      For the experiments presented in this study, we report in **Table 1** the particle growth rates (GR) determined from the

nSEMS size distributions. The growth rates in 3.2-8 nm and 5-15 nm were calculated using the 50% appearance time method described in Stolzenburg et al. (2018). From the calculated values in **Table 1**, we observe that $GR_{3.2-8 \text{ nm}}$ for the α-pinene + isoprene system (*αIP-30,20*) at the first concentration stage is around 18 nm h$^{-1}$ compared to ~ 77 nm h$^{-1}$ for the α-pinene only system (*α-30,20*). This is a factor of ~ 4 difference. While $GR_{5-15 \text{ nm}}$ represents a factor of 2 to 3 difference between *αIP-30,20* compared to *α-30,20*. From these values, one would conclude that isoprene does not contribute to the growth in the size range

reported here. Nevertheless, by looking at the aerosol mass concentration (see **Fig. S3** in the supplement), the mass reached during the experiment *αIP-30,20* is identical in the presence and absence of isoprene at -30 ℃ and 20 % RH. Reaching the same mass with a lower number of particles for the experiment with isoprene (*αIP-30,20*) compared to *α-30,20*, means that the growth rates at larger sizes (> 15 nm) are higher in the presence of isoprene. This is consistent with the fact that the particle size reached in the presence of isoprene is higher. Most likely, isoprene might enhance growth at higher sizes (> 15 nm) in this

study.

### 3.2.2 Influence of relative humidity on α-pinene system at -50 °C

**Figure 4** shows mass defect plots for the pure α-pinene experiments at -50 °C at low and high relative humidity: the gas and particle phase of α-pinene at -50 °C, 20 % RH (**Fig. 4a** and **Fig. 4d**); and the gas and particle phase of α-pinene at -50 °C, 60 to 100 % RH (**Fig. 4b** and **4e**). In both gas and particle phase at high and low RH, we detected $C_{8-10}$ monomers and $C_{18-20}$

dimers. $C_{10}H_{16}O_{4-7}$ and $C_{20}H_{32}O_{5-11}$ are the most prominent signals (see **Fig. S2** in the supplement).

      The relative humidity change from 20 % to 60 - 100 % does not have a significant influence on the gas phase composition at temperatures of -50 °C (**Fig. 4c**), meaning that most of the gaseous compounds detected contribute practically equal to the total signal when the humidity changes over the reported range. In contrast, there are changes in the particle phase signal. Although, the intensity difference (**Fig. 4f**) does not show a clear humidity effect on the particle chemical composition;

however, this comparison is based on the normalized signal (contribution of every compound to the total intensity). When looking only at the total intensity in the particle phase, we do observe an increase by a factor of ~ 3 in the total signal for the system at high RH (*α-50,60-100*) compared with the system at low RH (*α-50,20*). This observation can be likely attributed to



the change on the mass distribution (see **Fig. S3** in the supplement), which indicates that at similar α-pinene and ozone mixing ratio, and the same temperature, the mass concentration increases possibly due to the effect of the relative humidity in the

system. Besides, a possible impact of relative humidity on particle viscosity can influence particle mass formed, studies by Grayson et al. (2016) and Galeazzo et al. (2021) have reported lower viscosity with higher SOA mass concentration along with RH-dependence of viscosity for organic particles.

Our findings are consistent with previous experiments. Saathoff et al. (2009) observed that humidity has a significant influence on α-pinene SOA yields for lower temperatures. Cocker III et al. (2001) reported that the yield of SOA at higher RH

for α-pinene ozonolysis (relative to the system at dry conditions) increases possibly due to the uptake of water. One explanation for this observation could be that the rate constant value of the α-pinene ozonolysis can be affected by the RH (Zhang et al., 2018). Nevertheless, our semi-continuous particle phase measurements do not allow to draw any conclusions on the magnitude of the rate constants. Continuous particle phase measurements under different RH conditions are required in order to better understand the RH effect on the SOA formation.

In general, for the experiments presented in this work, most of the compounds that are present in the gas phase are detected as well in the particle phase, although the relative contributions to the total signal can vary depending on the phase. The more oxygenated material in the gas phase, specifically for $C_{20}$ dimers with $n_O > 13$ is not observed in the particle phase. This is probably because of their very low concentrations and the difficulty to distinguish between real particle signal and background. We conjecture that especially at low temperatures this issue might be related to the fact that at lower temperatures,

the autooxidation process to form HOM is slower, therefore, the oxygen content and O:C decrease (Stolzenburg et al., 2018; Ye et al., 2019; Simon et al., 2020). The low contribution of these compounds in the gas phase might be reflected in the particle phase. Additionally, the heating cycle that evaporates all the particulate material collected on the filament can potentially result in the thermal decomposition of some of the larger molecular weight compounds. Therefore, it is possible that a break-up of some molecules occurs.

**3.3 Volatility distribution of particle phase compounds**

**Figure 5** shows the volatility distribution of the oxidation products in the particle phase measured by the TD-DMA for the experiments reported in this work (in linear scale **Fig. S4** in the supplement). The volatility calculation was done by using the parametrization introduced by Donahue et al. (2011), and modified by Stolzenburg et al. (2018) and Simon et al. (2020). It is expressed as the logarithm of the saturation mass concentration, $\log_{10} c_i^*$ in µg m$^{-3}$, from the number of carbon and oxygen

atoms in the specific molecules. This approximation parameterizes the volatility of a molecule based on its functional groups and, a free parameter to distinguish between monomers and dimers.

In **Fig. 5** each volatility bin contains the summed intensity of the oxidation products measured in the particle phase and it is normalized by the total signal. Most of the classes are distributed over the range of the volatility values that are displayed and at lower temperatures, lower volatilities are observed (experiments *α-50,20* and *α-50,60-100*). This observation

is due to the fact of the strong dependency between saturation concentration and temperature. Essentially, there are no





significant differences between the experiments at -30 °C (*α-30,20* compared with *αIP-30,20*) or between the experiments at -50 °C (*α-50,20* compared with *α-50,60-100*), which indicates that temperature is the main parameter affecting the volatility distribution for the experiments reported here.

According to the volatility regimes proposed by Donahue et al. (2012) and Schervish and Donahue (2020), the particle

phase detected compounds correspond mainly to Low Volatility Organic Compounds (LVOC) and Extremely Low Volatility Compounds (ELVOC) and Ultralow Volatility Organic Compounds (ULVOC). With this parametrization we are able to approximate the saturation mass concentration for the particle phase compounds measured using the TD-DMA in the CLOUD chamber. For this parametrization we assume that the elemental composition is one of the main parameters to take into account.

### 3.4 Nucleation rates as a function of the total HOM

Previous CLOUD studies have reported nucleation rates ($J_{1.7nm}$) as a function of the total HOM concentration from α-pinene oxidation for different temperatures and gas mixtures (Kirkby et al., 2016; Heinritzi et al., 2020; Simon et al., 2020). For the experiments discussed in the present study the new particle formation rates have not been reported yet. For this reason, **Table 1** gives an overview of the experimental conditions for the experiments *α-30,20*, *αIP-30,20*, *α-50,20* and *α-50,60-100*; it further includes the HOM total concentration and derived $J_{1.7nm}$ from the PSM data (see method description in Section 2.4).

### 3.4.1 New Particle formation on pure α-pinene experiments

**Figure 6** displays pure biogenic $J_{1.7nm}$ vs total HOM concentration at different temperatures for pure α-pinene (Simon et al., 2020), in which can be seen that the total HOM concentration and their nucleation rates have a strong dependence on the temperature. As the temperature decreases, the nucleation rates increase strongly for a given HOM concentration. In other terms, the total HOM concentration needed to reach the same nucleation rate can be up to 2 orders of magnitude higher for

+25 °C compared to -50 °C. As described by Simon et al. (2020) this can be attributed to the reduction in volatility with decreasing temperature. In other words, at low temperatures, molecules with less oxygen content can lead to the same nucleation rate as more highly oxygenated molecules at higher temperatures. Additionally, **Fig. 6** includes the data points at -30 °C and -50 °C from pure α-pinene experiments reported in this study (*α-30,20*, *α-50,20* and *α-50,60-100*). However, it can be observed that they do not follow the trend at their corresponding temperature.

For the pure α-pinene systems (*α-30,20*, *α-50,20* and *α-50,60-100*) and complementary pure α-pinene experiments at +5 °C and at -10 °C, we have calculated the HOM yield as described in Simon et al. (2020) and found that the resulting values are higher than previously reported (see **Fig. S5** in the supplement). In order to investigate a possible reason for this finding, we have chosen two representative experiments at -10 °C and 80 to 90 % RH with different levels of α-pinene and ozone. **Fig. 7** shows the mass defect plots for the gas phase chemical composition of the oxidation products. In one experiment (**Fig. 7a**)

α-pinene and the ozone mixing ratio were between 0.2 to 0.8 ppbv and 40 to 50 ppbv, respectively, while for the second experiment (**Fig. 7b**) the mixing ratios were 2 to 3 ppbv and 100 ppbv, respectively. From **Fig. 7c** it can be concluded that the formation of HOM with low oxygen content is favored when the α-pinene and ozone mixing ratio are higher (relative to the





system at low levels of precursor gases). An explanation for this is that the high concentration of RO$_2$ enhances the terminating reactions before the autoxidation can lead to high oxygen content for the products. As the compounds with low oxygen content tend to have higher saturation vapor pressures, they do not contribute efficiently to new particle formation. For this reason, a given total HOM concentration is not unambiguously tied to a new particle formation rate (even at constant temperature). The magnitude of the precursor gas mixing ratio (more specifically the full volatility distribution of the products and not just the simple measure of total HOM) also needs to be taken into account (see **Fig.S6** in the supplement). In summary, the lower $J_{1.7nm}$ values compared with previous studies are very likely due to the higher α-pinene and ozone mixing ratios used in the present study. There are several compounds with low oxygen content that contribute to the total HOM concentration in the gas phase while these do not contribute to the formation of new particles.

### 3.4.2 The influence of isoprene on new particle formation

In order to make the present study comparable with other studies that reported a suppression effect of isoprene on biogenic new particle formation, the values of the isoprene-to-monoterpene carbon ratio ($R$) are also provided in Table 1, here and in previous studies $R$ is essentially the ratio between isoprene and α-pinene; for experiment *αIP-30,20*, R equals to 14.4 and 6.1 (for two steady-state periods in *αIP-30,20*).

Fig. 8 shows pure biogenic nucleation rates at 1.7 nm against total HOM concentration at different temperatures for the α-pinene and α-pinene + isoprene systems (Kirkby et al., 2016; Heinritzi et al., 2020; Simon et al., 2020). How rapidly particles are formed in a pure biogenic system depends strongly on the temperature and on the ion conditions. In general, we observe increasing nucleation rates at lower temperatures and at GCR conditions. The presence of isoprene lowers the nucleation rate (relative to the pure α-pinene system at similar conditions); this is known as isoprene suppression of new particle formation. In this regard, there is a suppression on the new particle formation caused by adding isoprene on an α-pinene system at -30 ºC and 20 % RH. However, it has been reported that the suppression effect is stronger when α-pinene is lower (and $R$ is higher, see **Fig. S7** in the supplement). For instance, for a plant chamber experiment that R = 19.5 resulted in no significant new particle formation (Kiendler-Scharr et al., 2009). Additionally, in the Michigan forest with R = 26.4, NPF events did not occur frequently (Kanawade et al., 2011). In spite of that, one has to consider that the suppression effect at a given value of $R$ likely decreases as temperature decreases and so does the saturation vapor pressure of the oxidation products.

### 4 Conclusions

In this study, we showed the capability of the Thermal Desorption-Differential Mobility Analyzer (TD-DMA) coupled to a chemical ionization time-of-flight mass spectrometer for measuring HOMs in newly formed nano aerosol particles. Together with the nitrate CI-APi-TOF mass spectrometer, this set up is capable of measuring gas and particle phase, allowing a direct comparison as both measurements use the identical chemical ionization and detector.



For the pure biogenic NPF experiments performed at -50 °C and -30 °C in the CLOUD chamber at CERN, we detected in the particle phase (diameter up to ~ 100 nm) compounds such as $C_{10}H_{16}O_{3-9}$, and $C_{20}H_{32}O_{5-13}$. Especially for the system with

isoprene added, $C_5$ ($C_5H_{10}O_{5-7}$) and $C_{15}$ compounds ($C_{15}H_{24}O_{5-10}$) can be an important fingerprint to identify secondary organic aerosol from this biogenic source. Based on the elemental composition, we calculated the saturation mass concentration, and according to the volatility regimes, the particle phase compounds correspond mainly to Low Volatility Organic Compounds (LVOC) and Extremely Low Volatility Compounds (ELVOC) and Ultralow Volatility Organic Compounds (ULVOC).

    We also showed that at -30 °C and an isoprene-to-monoterpene carbon ratio R = 14.4 and 6.1, there is a reduction of

the nucleation rate (compared to the pure α-pinene system at similar conditions). In this way, isoprene suppresses NPF at -30 °C. Nevertheless, this suppression effect can be stronger at higher temperatures and at high *R*.

    Lastly, the lower $J_{1.7}$ values compared with previous studies are very likely due to the higher α-pinene and ozone mixing ratios used in the present study. There are several compounds with low oxygen content that contribute to the total HOM concentration in the gas phase while these do not contribute to the formation of new particles. For this reason, a given total

HOM concentration is not unambiguously tied to a new particle formation rate (even at constant temperature). The magnitude of the precursor gas mixing ratio, and thus the full volatility distribution, also needs to be taken into account.

*Data availability.* Data related to this article are available upon request to the corresponding authors.

*Supplement.* The supplement related to this article is available online at:

*Author contributions.* L. C., B. R., G. M., M. Sim., A. C. W., T. M., M. G., F. A., R. B., B. B., Z. B., R. C., B. C., J. Du., H. F., L. G. C., X.-C. H., V. H., W. K., H. L., C. P. L., B. L., N. G. A. M., V. M., H. E. M., R. M., R. L. M., B. M., U. M., A. O., J. P., M. P., A. A. P., W. S., B. S., J. S., D. S., Y. S., M. S., Y. J. T., P. T., A. T., S.V., M. W., D. S. W., S. K. W., A. W., W.

Y., W. Y., M. Z.-W., U. B., I. E.-H., R. C. F., K. H., J. Kir., M. K., K. L., O. M., R. V., P. M. W., A. K., and J. Cu. prepared the CLOUD facility and measurement instruments. L. C., B. R., G. M., A. C. W., T. M., M. G., A. A., F. A., R. B., B. B., J. Du., L. G. C., V. H., H. L., C. P. L., N. G. A. M., V. M., R. M., D. M., R. L. M., B. M., U. M., W. S., B. S., D. S., M. S., C. T., Y. J. T., A. T., D. S. W., S. K. W., M. Z.-W., I. E.-H., J. Kir., R. V., and P. M. W. collected the data. L. C., B. R., M. H., G. M., F. A., L. D., R. L. M., U. M., W. S., B. S., S. K. W., and R. C. F. analyzed the data. L. C., M. H., M. Sim., A. C. W.,

T. M., M. G., L. D., R. L. M., U. M., M. S., W. S., D. S., U. B., I. E.-H., R. C. F., A. H., K. H., J. Kir., M. K., O. M., H. S., N. M. D., A. K., and J. Cu. contributed to the scientific discussion and interpretation of the results. L. C., M. H., A. C. W., L. D., U. B., R. C. F., H. S., N. M. D., A. K., and J. Cu. contributed to writing the manuscript.

*Competing interests.* The authors declare that they have no conflict of interest.




*Acknowledgments*. We thank CERN for providing the CLOUD facility to perform the experiments and the CLOUD community for supporting this study. We especially would like to thank Katja Ivanova, Timo Keber, Frank Malkemper, Robert Sitals, Hanna Elina Manninen, Antti Onnela, and Robert Kristic for their contributions to the experiment.

*Financial support*. This work was supported by Innovative Training Networks – ITN (CLOUD-Motion H2020-MSCA-ITN-2017 no. 764991) and by the German Ministry of Science and Education (CLOUD-16, 01LK1601A). US National Science Foundation Award (AGS-1801280, AGS-1801574, AGS-1801897). Swiss National Science Foundation (20020_172602, BSSGI0_155846).

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



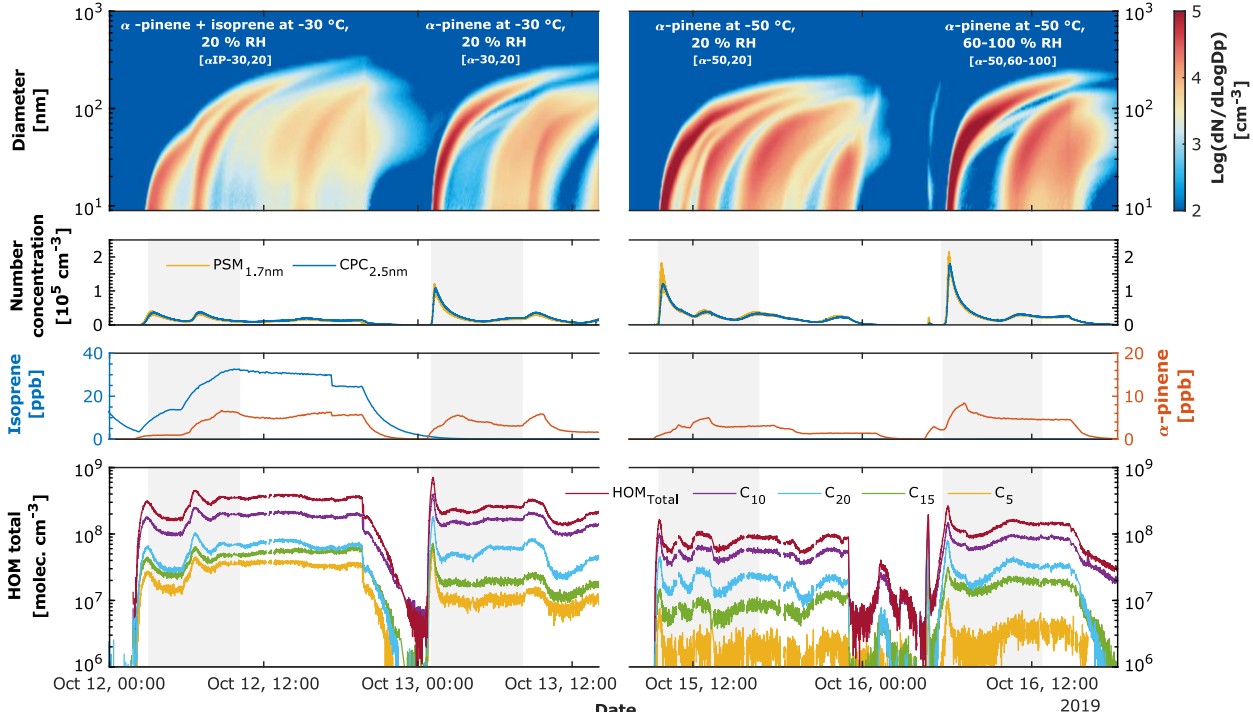

**Figure 1. Experimental overview for pure biogenic new particle formation. First panel: particle size distribution for four different experiments: α-pinene + isoprene at -30 °C and 20 % RH (αIP-30,20); α-pinene at -30 °C and 20 % RH (α-30,20); α-pinene at -50 °C and 20 % RH (α-50,20) and α-pinene at -50 °C and 60-100 % RH (α-50,60-100). The color scale represents the log 10 of the normalized particle concentration in cm⁻³. Second panel: Particle number concentration in cm⁻³ measured by the PSM with a cut-off diameter of 1.7 nm and CPC 2.5 nm. Third panel: Mixing ratio in ppbv for the biogenic precursor gases, isoprene and α-pinene. Fourth panel: Evolution of total HOM concentration in molec. cm⁻³, measured in the gas phase by the Nitrate CI-APi-TOF mass spectrometer. HOM total is defined as the sum of $C_5$, $C_{10}$, $C_{15}$ and $C_{20}$ carbon classes which are shown as well. Ozone level is not shown, though remains stable over the whole period ~ 100 ppbv. The shaded areas refer to the time where the particles were collected using the TD-DMA.**

**Table 1. Summary of the main parameters for four pure biogenic new particle formation experiments.**

| Experiment | Isoprene [ppb] | α-pinene [ppb] | Isoprene-to-monoterpene carbon ratio (R) | Ozone [ppb] | T [°C] | RH [%] | HOM total* [molec. cm⁻³] | $J_{1.7}$* [cm⁻³ s⁻¹] | Growth rate 3.2-8 nm [nm h⁻¹] | Growth rate 5-15 nm [nm h⁻¹] |
|---|---|---|---|---|---|---|---|---|---|---|
| αIP-30,20 | 13.71 | 0.95 | 14.4 | 98.58 | -30 | 20 | 1.50e8 | 7.29 | 18.0 | 22.8 |
| | 31.38 | 5.12 | 6.1 | 101.56 | -30 | 20 | 3.04e8 | 10.10 | NA | 39.0 |
| α-30,20 | ~ 0 | 3.35 | NA | 102.10 | -30 | 20 | 2.20e8 | 23.76 | 76.9 | 77.1 |
| α-50,20 | ~ 0 | 3.04 | NA | 100.55 | -50 | 20 | 6.72e7 | 51.24 | 41.1 | 42.0 |
| α-50,60-100 | ~ 0 | 7.72 | NA | 110.20 | -50 | 60-100 | 8.00e7 | 79.17 | 63.4 | 78.4 |

\* Run-to-run experimental uncertainties of HOMs is ± 20 % and for $J_{1.7}$ is ± 30 %. NA, Not Applicable.



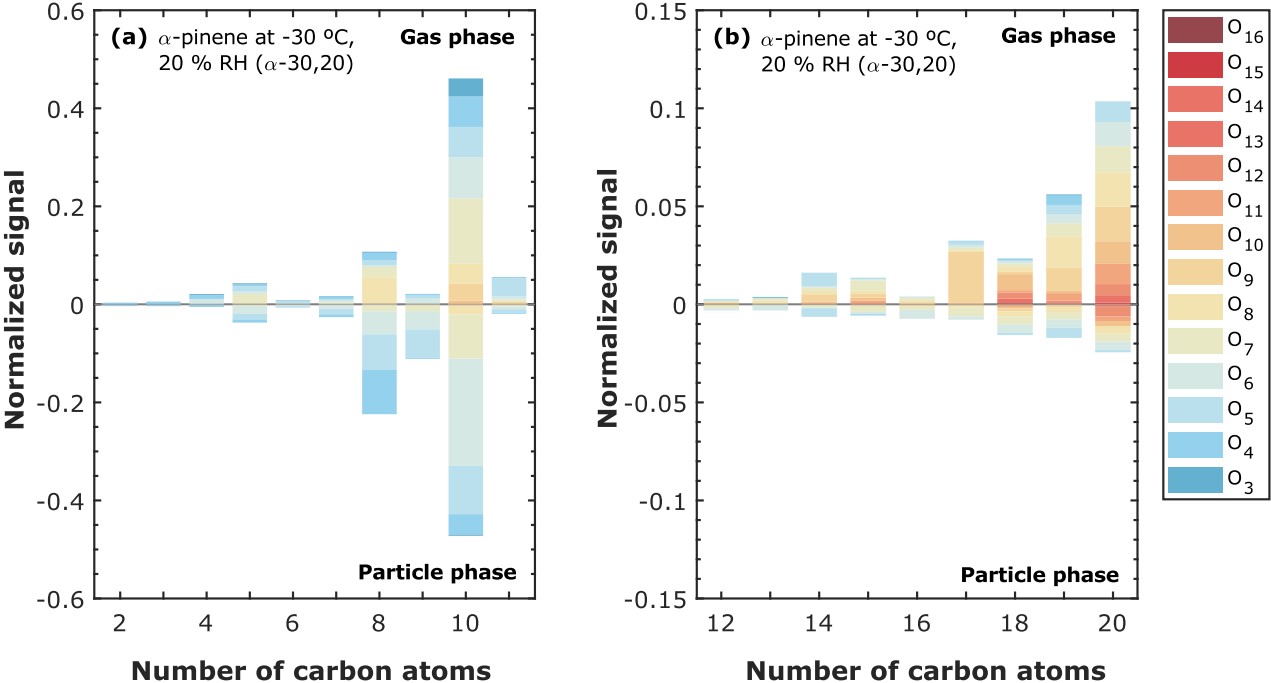

**Figure 2. Carbon atom distribution and oxygen atom content in gas and particle phase molecules for α-pinene oxidation products at -30 °C and 20 % RH (α-30,20). Both phases are measured with a Nitrate CI-APi-TOF Mass Spectrometer, while the TD-DMA is coupled to it for particle phase measurements. (a) carbon atom distribution $C_{2-11}$ and (b) carbon atom distribution $C_{12-20}$. The level of α-pinene was between 1 and 8 ppbv and Ozone level was stable at ~ 100 ppbv. The intensities are normalized by the total signal in each system. Each color represents a specific number of oxygen atoms in the range of 3 to 16.**






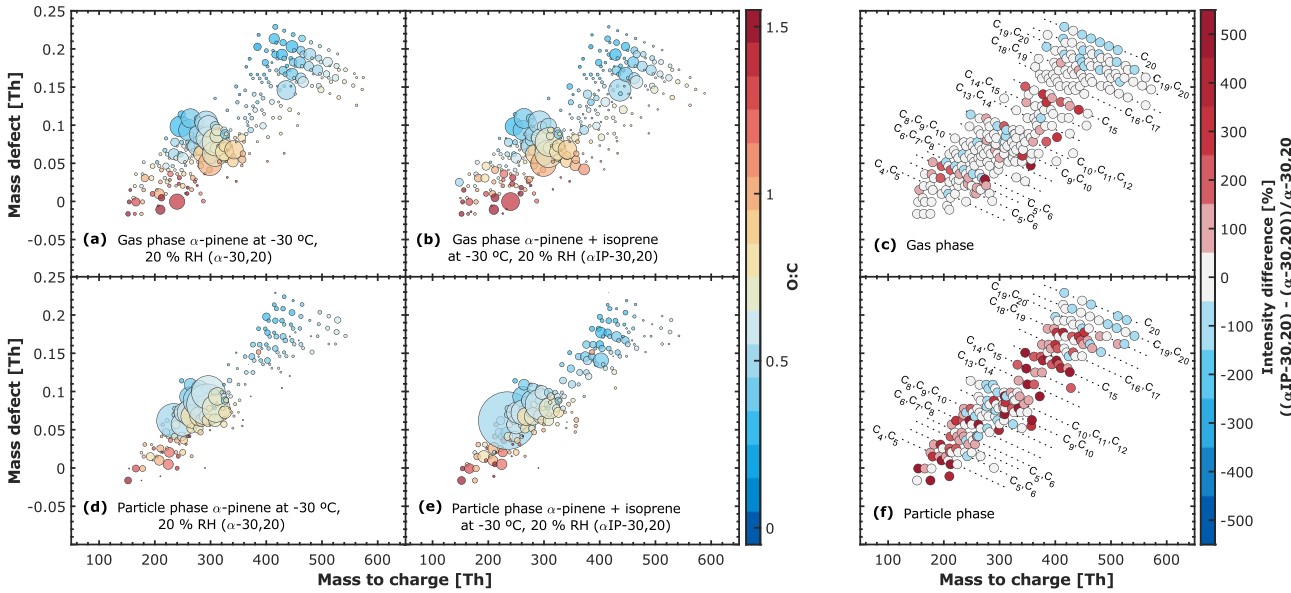


**Figure 3. Mass defect plots of gas and particle phase and the intensity difference between them. Both phases are measured with a Nitrate CI-APi-TOF Mass Spectrometer, while the TD-DMA is coupled to it for particle phase measurements. (a) gas and (d) particle for α-pinene oxidation products at -30 °C and 20 % RH (α-30,20). (b) gas and (e) particle for α-pinene + isoprene oxidation products at -30 °C and 20 % RH (αIP-30,20). The level of α-pinene was between 1 and 8 ppbv in both experiments, while isoprene was present only in experiment αIP-30,20 reaching up to 30 ppbv. Ozone levels were ~ 100 ppbv in both experiments. The symbol sizes in (a), (b), (d) and (e) are the intensities normalized by the total signal in each system. The intensity difference in gas (c) and in particle (f) is indicated as ((αIP-30,20) - (α-30,20))/ α-30,20. The color scale represents the difference in percent.**




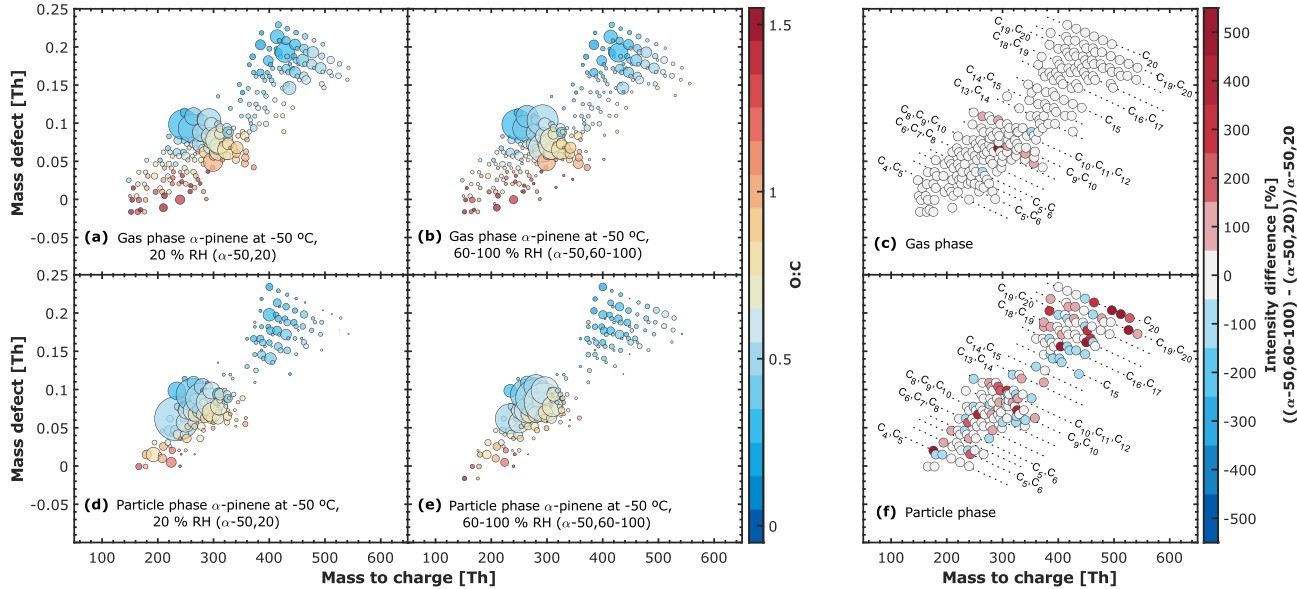

**Figure 4. Mass defect plots of gas and particle phase and the intensity difference between them. Both phases are measured with a Nitrate CI-APi-TOF Mass Spectrometer, while the TD-DMA is coupled to it for particle phase measurements. (a) gas and (d) particle for α-pinene oxidation products at -50 °C and 20 % RH (α-50,20). (b) gas and (e) particle for α-pinene oxidation products at -50 °C and 60 to 100 % RH (α-50,60-100). The level of α-pinene was between 1 and 8 ppbv and Ozone levels were ~ 100 ppbv in both experiments. The symbol sizes in (a), (b), (d) and (e) are the intensities normalized by the total signal in each system. The intensity difference in gas (c) and in particle (f) is indicated as ((α-50,60-100) - (α-50,20))/ α-50,20. The color scale represents the difference in percent.**

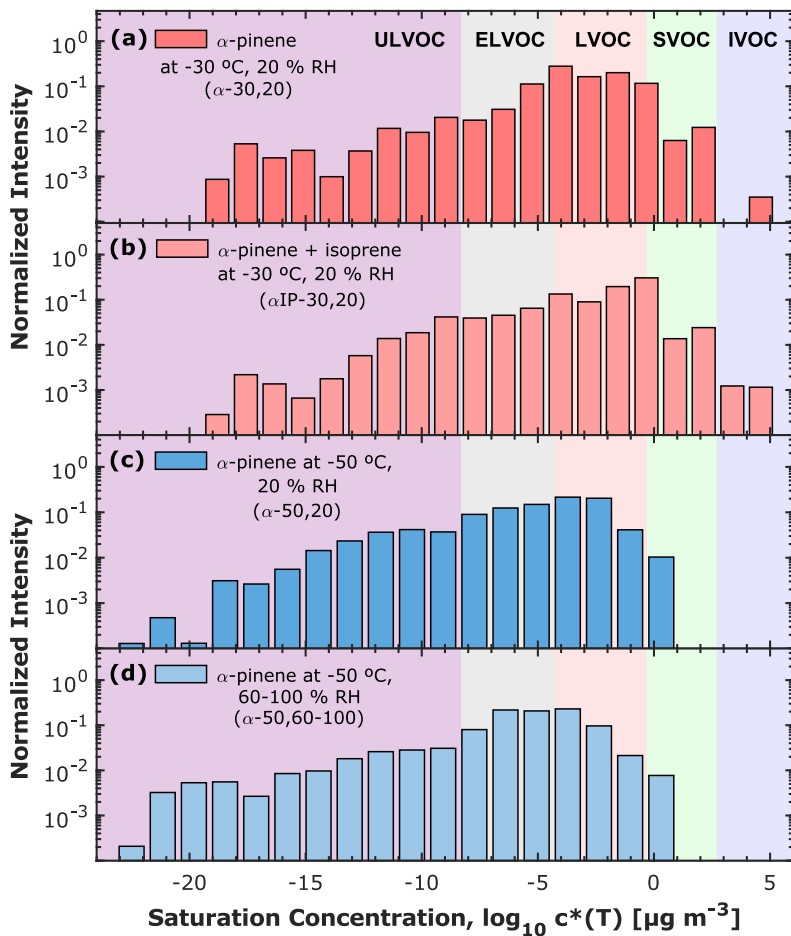

705

**Figure 5. TD-DMA Volatility distribution of the measured oxidation products in the particle phase for four different experiments: (a) α-pinene at -30 °C and 20 % RH (α-30,20); (b) α-pinene + isoprene at -30 °C and 20 % RH (αIP-30,20); (c) α-pinene at -50 °C and 20 % RH (α-50,20) and (d) α-pinene at -50 °C and 60-100 % RH (α-50,60-100). Every individual volatility bin includes the sum of the intensity for the oxidation products normalized by the total signal in each system. Every individual volatility bin is defined at 300 K, shifted, and widened according to their corresponding temperature. The color bands in the background indicate the volatility regimes as in Donahue et al. (2012) and in Schervish and Donahue (2020). The normalized intensity is dimensionless. Nevertheless, it should be noted that the particle phase signal is given in normalized counts per second integrated over the evaporation time [ncps. s].**

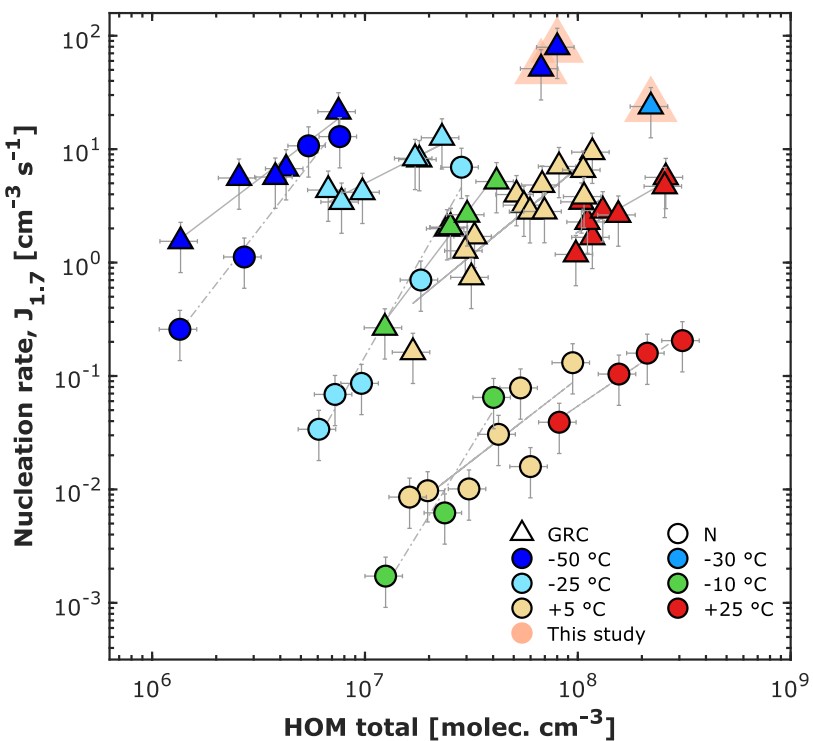

**Figure 6. Pure biogenic nucleation rates of pure α-pinene at 1.7 nm diameter against total HOM concentration at different temperatures. HOM total is defined as the sum of C₅, C₁₀, C₁₅ and C₂₀ carbon classes. Triangles represent galactic cosmic rays (GCR) conditions and circles represent Neutral conditions. Data points at -50 °C, -25 °C, -10 °C, +5 °C and +25 °C are from Simon et al. (2020). The points with orange marker on the background are the contribution of this study (experiments α-30,20, α-50,20 and α-50,60-100). Solid and dashed lines represent power-law fits to GCR and Neutral conditions. Bars indicate 1α run-to-run uncertainty. The overall systematic scale uncertainty of HOM of +78 % and -68 % and is not shown.**



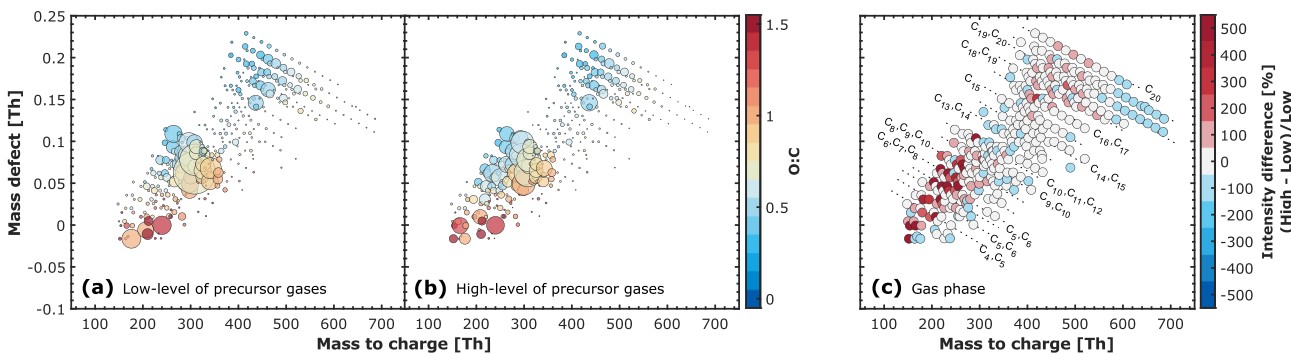

**Figure 7. Mass defect plots of gas phase for two different systems and the normalized difference between them. Gas phase is measured with a Nitrate CI-APi-TOF Mass Spectrometer, (a) α-pinene oxidation products at -10 °C and 90 % RH at low-level of precursor gases, (b) α-pinene oxidation products at -10 °C and 80 % RH at high-level of precursor gases. α-pinene was 0.2 - 0.8 ppbv and Ozone ~ 40 - 50 ppbv in (a), and α-pinene was 2 - 3 ppbv and Ozone ~ 100 ppbv in (b). The symbol sizes and colors in (a) and (b) represent the intensities normalized by the total signal in each system. (c) The difference between the normalized signals shown in (a) and (b) is represented by the color scale.**



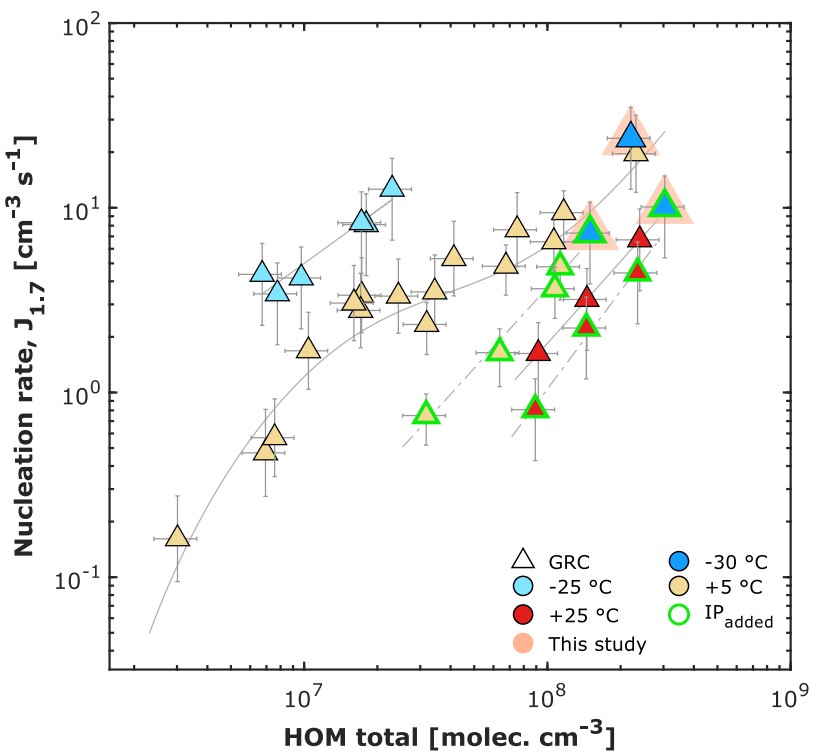

**Figure 8. Pure biogenic nucleation rates at 1.7 nm diameter against total HOM concentration at different temperatures for α-pinene and α-pinene + isoprene systems. HOM total is defined as the sum of C5, C10, C15 and C20 carbon classes. Triangles represent Galactic Comic Rays (GCR) conditions. Data points at +5 °C and +25 °C are from Kirkby et al. (2016) and Heinritzi et al., (2020).** **Data points at -25 °C are from Simon et al., (2020). The value of isoprene-to-monoterpene carbon ratio (R) varies from 1.5 to 6.5 for Heinritzi et al., (2020), and R equals to 14.4 and 6.1 for this study. The points with orange marker on the background indicate the contribution of this work. Solid lines represent power-law fits to GCR conditions of the systems with α-pinene only and dashed lines are the power-law fits of the systems with isoprene added. Bars indicate 1α run-to-run uncertainty. The overall systematic scale uncertainty of HOM of +78 % and -68 % and is not shown.**