# Peer review of "Chemical composition of nanoparticles from $\alpha$ -pinene nucleation and the influence of isoprene and relative humidity at low temperature"

_Atmospheric Chemistry and Physics, 2021_

## Author Comment (AC1)

Interactive discussion on *Chemical composition of nanoparticles from α-pinene nucleation and the influence of isoprene and relative humidity at low temperature* by Caudillo et al., Atmos. Chem. Phys. Discuss., https://doi.org/10.5194/acp-2021-512

We thank the referees for their positive and constructive feedback. We greatly appreciate their time investment for carefully reviewing our manuscript. We have given different font colors for facilitating the discussion. The specific and technical comments addressed by the referees are in black, while our answers are in blue. Any changes made to the manuscript can be found in red.

**Anonymous Referee #1**

Caudillo and co-authors present and discuss the gas and particle phase composition of pure biogenic nucleation events measured with a nitrate chemical ionization atmospheric pressure interface time of flight mass spectrometer (coupled with a thermal desorption differential mobility analyzer for the particle phase) in the CLOUD chamber at a range of conditions representing free tropospheric conditions. Specifically, alpha-pinene and a mix of alpha-pinene and isoprene were oxidized at -30 deg C and - 50 deg C, and at 20 % or 60 - 100 % relative humidity. The authors find C8-10 monomers and C18-20 dimers as major compounds, and C5 and C15 compounds contributing to particle growth when isoprene is present in the system. I very much appreciate the systematic analysis. The manuscript is well-written and the experimental results are thoroughly discussed. In my opinion, this is an original and valuable contribution to the field. Therefore, it should be published in ACP after minor revisions.

Specific comments:

In the abstract, the last sentence ("Besides the chemical information...", lines 64-66) was confusing to me. After reading section 3.4.1, I suggest to be more specific in the abstract, e.g., "Compared with previous studies, we found lower nucleation rates measured at 1.7 nm, very likely due to higher alpha-pinene and ozone mixing ratios used in the present study."

The sentences have been modified as suggested:

For the experiments reported here, most likely isoprene oxidation products enhance the growth of particles larger than 15 nm. Additionally, we report on the nucleation rates measured at 1.7 nm ($J_{1.7nm}$) and compared with previous studies, we found lower $J_{1.7nm}$ values, very likely due to the higher α-pinene and ozone mixing ratios used in the present study.

In section 2.1, there is no information about GCR conditions during the experiments, please add.

The requested information about GCR conditions has been added accordingly in Section 2.1.

In order to promote particle production from ions, Galactic Cosmic Rays conditions (GCR) can be achieved by turning off the high voltage field (30 kV m$^{-1}$). The equilibrium ion-pair concentration in the chamber due to GCR is around 700 cm$^{-3}$ (Kirkby et al., 2016).

The experiments relevant for this work were done under GCR conditions and in a flow-through mode with continuous addition of the reactants, performed at -50 °C and -30 °C, at low and high relative humidity to simulate pure biogenic new particle formation at a range of free tropospheric conditions.

In section 2.2, please add some more information about the heating procedure of the filament. Is the temperature slowly ramped up, or do you apply high temperature directly to instantaneously vaporize the sample? This is also relevant for the discussion of potential thermal decomposition of molecules in section 3.2.2.

The information regarding the heating produce of the filament was added in the Section 2.2.

For the experiments that are reported in this work, a filament of platinum/rhodium (90:10) was used, and an integral, non-size selective mode of operation was chosen in order to maximize the mass of collected particles. For desorbing the sample, an electric current was applied to the filament and ramped linearly over a duration of approximately 1 minute. Due to the very low experimental temperatures, cold sheath flows and isolated inlet lines were installed in order to avoid drastic temperature changes between the CLOUD chamber and the instrument. Evaporation of particulate material before the active heating should therefore not be substantial.

Regarding the non-size selective mode of operation of the TD-DMA, it would be helpful to get an idea about the contribution of freshly nucleated particles vs. grown particles to the sampled mass. From the measured particle size distributions and the PSM and CPC total number concentrations, could you calculate a rough estimate of the volume/mass fraction of particles < 15 nm in the samples collected in the periods shown in Figure 1 as shaded areas? Please add this information to Table 1.

The calculation was done accordingly and the requested information has been added to Table 1.

For calculating the mass fraction of particles collected on the filament during the TD-DMA collection time we used only SMPS size distributions, in a range from 9 nm to 834 nm. We estimated that the mass of particles with sizes smaller than 9 nm is negligible (by comparing to nanoSMPS data in a size range from 2 to 63 nm). Thus, the mass fraction reported in Table 1 takes only into account SMPS measurements. From these data we derived the mass fractions for the collected particles with a diameter smaller than 15 nm (Mass $_{< 15\,nm}$) and for a diameter larger than 15 nm (Mass $_{> 15\,nm}$).

This calculation is an ideal estimate, it does not take into account uncertainties due to charging efficiency, collection efficiency and transmission efficiency in the TD-DMA as reported in detail by Wagner et al. (2018) for the size-selective mode.

Table 1. Summary of the main parameters for four pure biogenic new particle formation experiments.

| Experiment | Isoprene [ppb] | α-pinene [ppb] | Isoprene-to-monoterpene carbon ratio (R) | Ozone [ppb] | T [°C] | RH [%] | HOM total* [molec. cm⁻³] | $J_{1.7}$* [cm⁻³ s⁻¹] | Growth rate 3.2-8 nm [nm h⁻¹] | Growth rate 5-15 nm [nm h⁻¹] | Mass $_{< 15\,nm}$ /Mass $_{> 15\,nm}$** [%] |
|---|---|---|---|---|---|---|---|---|---|---|---|
| αIP-30,20 | 13.71 | 0.95 | 14.4 | 98.58 | -30 | 20 | 1.50e8 | 7.29 | 18.0 | 22.8 | 0.29 / 99.7 |
|  | 31.38 | 5.12 | 6.1 | 101.56 | -30 | 20 | 3.04e8 | 10.10 | NA | 39.0 |  |
| α-30,20 | ~ 0 | 3.35 | NA | 102.10 | -30 | 20 | 2.20e8 | 23.76 | 76.9 | 77.1 | 0.11 / 99.9 |
| α-50,20 | ~ 0 | 3.04 | NA | 100.55 | -50 | 20 | 6.72e7 | 51.24 | 41.1 | 42.0 | 0.26 / 99.7 |
| α-50,60 | ~ 0 | 7.72 | NA | 110.20 | -50 | 60 | 8.00e7 | 79.17 | 63.4 | 78.4 | 0.09 / 99.9 |

* Run-to-run experimental uncertainties of HOMs is ± 20 % and for $J_{1.7}$ is ± 30 %. NA, Not Applicable. **Mass fraction of particles collected on the filament during the TD-DMA collection time, calculation based on SMPS mass distributions.

In Figure 3f, to me it is not obvious that specifically C4-5 and C13-16 compounds are enhanced as stated in lines 237/238. Please clarify.

Many thanks for pointing this out. Regarding the isoprene effect, we do observe an increase in the intensity for several species in the particle phase, not only for $C_5$ and $C_{15}$ compounds. We believe that this might be related to the method we apply for evaporating the particles (thermal desorption), possibly fragmentation and chemical reactions are occurring. We are further investigating this phenomenon in a forthcoming study.

We clarified in Section 3.2.1 that $C_5$ and $C_{15}$ compounds correspond to $C_5H_{10}O_{5-7}$ and $C_{15}H_{24}O_{5-10}$ which are observed in both gas and particle phase.

Fig. 3f shows the intensity difference in the particle phase. In contrast with what is observed in the gas phase, the particle phase effects seem more diverse. There is an increase in the intensity difference for several species in the system with isoprene (*αIP-30,20*), such as $C_{4-5}$, $C_{13-16}$ and some $C_{17-19}$ (see Fig. S1 in the supplement). Especially, a distinct group of $C_{15}$ compounds $C_{15}H_{24}O_{5-10}$ and $C_5H_{10}O_{5-7}$ can be identified in the particle and in the gas phase. A previous study has shown that isoprene can enhance particle growth rates despite its negative effect on nucleation (Heinritzi et al., 2020). The identification of $C_{15}$ dimers in nanometer-sized particles in the present study confirms this with a direct measurement. The suppressing effect of isoprene on nucleation will further be discussed in Section 3.4.2. However, isoprene can still contribute to the growth of particles by $C_5$ or by $C_{15}$ compounds. Additionally, these species can be an important fingerprint to identify SOA formation from a mixture of biogenic vapors containing isoprene.

In section 3.3, looking at Figures 5 and S4 I agree with the statement that mainly LVOC and ELVOC compounds were detected in the particle phase, however, the ULVOC compounds appear to be a minor fraction.

We agree as well that ULVOC do not represent the highest fraction. We have calculated the volatility bin fractions and reported these values in Table 2.

Table 2. Bin volatility fractions for the different experiments.

| Experiment | T [°C] | RH [%] | ULVOC [%] | ELVOC [%] | LVOC [%] | SVOC [%] | IVOC [%] |
|---|---|---|---|---|---|---|---|
| αIP-30,20 | -30 | 20 | 8.6 | 28.3 | 59.1 | 3.8 | 0.2 |
| α-30,20 | -30 | 20 | 5.9 | 44.1 | 48.1 | 1.9 | 0 |
| α-50,20 | -50 | 20 | 16.5 | 36.4 | 46.1 | 1.0 | 0 |
| α-50,60 | -50 | 60 | 13.8 | 50.6 | 34.8 | 0.8 | 0 |

In Section 3.3, we have mentioned that LVOC and ELVOC represents the major fractions, and ULVOC only a minor fraction.

We classified the volatility bins according to the regimes proposed by Donahue et al. (2012) and Schervish and Donahue (2020), and calculated the corresponding fractions (Table 2). Overall, the particle phase detected compounds correspond mainly to Low Volatility Organic Compounds (LVOC) and Extremely Low Volatility Compounds (ELVOC) by explaining more than 80 % of the signals, while Ultralow Volatility Organic Compounds (ULVOC) represent only a small fraction (between 6 and 17 %). With this parametrization we are able to approximate the saturation mass concentration for the particle phase

compounds measured using the TD-DMA in the CLOUD chamber. For this parametrization we assume that the elemental composition is one of the main parameters to take into account.

Section 3.4.2: Regarding the discussion of isoprene suppression of new particle formation, please consider adding a reference to Lee et al. (2016), doi:10.1002/2016JD024844.

The reference has been added accordingly in Section 3.4.2.

For instance, in a plant chamber experiment R = 19.5 resulted in no significant new particle formation (Kiendler-Scharr et al., 2009). Additionally, in the Michigan forest with R = 26.4, NPF events did not occur frequently (Kanawade et al., 2011). Lee et al. (2016) reported observations of NPF suppression in a rural forest in Alabama where R = 2.0. However, one has to consider that the suppression effect at a given value of *R* likely decreases as temperature decreases and so does the saturation vapor pressure of the oxidation products.

In Figures 6 and 8, please explain the meaning of "1α run-to-run uncertainty".

1σ run-to-run uncertainty refers to the experimental errors calculated based on run-to-run repeatability in the CLOUD chamber (experiments under identical conditions). Detailed information about this calculation has been previously published in Kirkby et al. (2016), Lehtipalo et al. (2018) and, Dada et al. (2020).

We have changed "1α run-to-run uncertainty" for "1σ run-to-run experimental uncertainty" in the caption for Figures 6 and 8.

Figure 6. Pure biogenic nucleation rates of pure α-pinene at 1.7 nm diameter against total HOM concentration at different temperatures. HOM total is defined as the sum of $C_5$, $C_{10}$, $C_{15}$ and $C_{20}$ carbon classes. Triangles represent Galactic Cosmic Rays (GCR) conditions and circles represent Neutral conditions. Data points at -50 °C, -25 °C, -10 °C, +5 °C and +25 °C are from Simon et al. (2020). The points with orange marker on the background are the contribution of this study (experiments α-30,20, α-50,20 and α-50,60). Solid and dashed lines represent power-law fits to GCR and Neutral conditions. Bars indicate 1 σ run-to-run experimental uncertainty. The overall systematic scale uncertainty of HOM of +78 % and -68 % is not shown.

Figure 8. Pure biogenic nucleation rates at 1.7 nm diameter against total HOM concentration at different temperatures for α-pinene and α-pinene + isoprene systems. HOM total is defined as the sum of $C_5$, $C_{10}$, $C_{15}$ and $C_{20}$ carbon classes. Triangles represent Galactic Cosmic Rays (GCR) conditions. Data points at +5 °C and +25 °C are from Kirkby et al. (2016) and Heinritzi et al., (2020). Data points at -25 °C are from Simon et al., (2020). The value of isoprene-to-monoterpene carbon ratio (R) varies from 1.5 to 6.5 for Heinritzi et al., (2020), and R equals to 14.4 and 6.1 for this study. The points with orange marker on the background indicate the contribution of this work. Solid lines represent power-law fits to GCR conditions of the systems with α-pinene only and dashed lines are the power-law fits of the systems with isoprene added. Bars indicate 1 σ run-to-run experimental uncertainty. The overall systematic scale uncertainty of HOM of +78 % and -68 % is not shown.

Technical comments:

lines 225, 301, 322: Make sure that there is no line break between sign and number in "-30" and "- 50".

We have attended this comment accordingly.

line 226: "While the gas..." is not a full sentence.

We have modified the sentence.

Figure 3 shows mass defect plots of gas and particle phase and the intensity difference between each phase at -30 °C. Figure 3a and Fig. 3d display the gas and particle composition of *α-30,20*, while the gas and particle composition of *αIP-30,20* are shown in Fig. 3b and Fig. 3e, respectively. As both phases were measured with the same instrument, they can be directly inter-compared.

Section 3.2, first paragraph: When presenting and discussing Figure 2, add a reference to supplementary material Figures S1 and S2 for other systems.

We have added this information as requested.

Figure 2 shows the carbon distribution as an overview of the compounds detected in gas and particle phase for a system where only α-pinene was oxidized (*α-30,20*). $C_{8-10}$ monomers (Fig. 2a) and $C_{18-20}$ (Fig. 2b) dimers are observed in the gas as well as in the particle phase. For instance, some of the signals with the highest intensity correspond to $C_{10}H_{16}O_{3-9}$, and $C_{20}H_{32}O_{5-13}$, especially $C_{10}H_{16}O_6$ and $C_{10}H_{16}O_7$ have an important presence in both phases. Overall, most of the compounds that are present in the gas phase are detected as well in the particle phase, although their relative contribution to the total signal can differ between the phases. The corresponding carbon distribution for the other systems can be found in Figures S1 and S2 in the supplement.

Line 248: "While GR..." is not a full sentence.

We have modified the sentence.

This is a factor of ~ 4 difference, while $GR_{5-15}$ nm represents a factor of 2 to 3 difference between *αIP-30,20* compared to *α-30,20*.

line 285: Change "autooxidation" to "autoxidation" to be consistent throughout the manuscript.

We have attended this comment accordingly.

lines 204, 312, 336, 375: Here, "new particle formation rate" is used, while J_1.7nm is introduced as "nucleation rate" in the manuscript. For consistency, I recommend using "nucleation rate" throughout the manuscript.

We have attended this comment accordingly.

In Figures 6 and 8: "GRC" should read "GCR" in the figure legend. Also, in the last sentence of the figure caption, remove "and" before "is not shown".

We have attended this comment accordingly.

[Figure]

Figure 6. Pure biogenic nucleation rates of pure α-pinene at 1.7 nm diameter against total HOM concentration at different temperatures. HOM total is defined as the sum of $C_5$, $C_{10}$, $C_{15}$ and $C_{20}$ carbon classes. Triangles represent Galactic Cosmic Rays (GCR) conditions and circles represent Neutral conditions. Data points at -50 °C, -25 °C, -10 °C, +5 °C and +25 °C are from Simon et al. (2020). The points with orange marker on the background are the contribution of this study (experiments α-30,20, α-50,20 and α-50,60). Solid and dashed lines represent power-law fits to GCR and Neutral conditions. Bars indicate 1 σ run-to-run experimental uncertainty. The overall systematic scale uncertainty of HOM of +78 % and -68 % is not shown.

[Figure]

Figure 8. Pure biogenic nucleation rates at 1.7 nm diameter against total HOM concentration at different temperatures for α-pinene and α-pinene + isoprene systems. HOM total is defined as the sum of $C_5$, $C_{10}$, $C_{15}$ and $C_{20}$ carbon classes. Triangles represent Galactic Cosmic Rays (GCR) conditions. Data points at +5 °C and +25 °C are from Kirkby et al. (2016) and Heinritzi et al., (2020). Data points at -25 °C are from Simon et al., (2020). The value of isoprene-to-monoterpene carbon ratio (R) varies from 1.5 to 6.5 for Heinritzi et al., (2020), and R equals to 14.4 and 6.1 for this study. The points with orange marker on the background indicate the contribution of this work. Solid lines represent power-law fits to GCR conditions of the systems with α-pinene only and dashed lines are the power-law fits of the systems with isoprene added. Bars indicate 1 σ run-to-run experimental uncertainty. The overall systematic scale uncertainty of HOM of +78 % and -68 % is not shown.

**Anonymous Referee #2**

General comments

Caudillo et al. present chamber measurements of new-particle formation from α-pinene (AP) oxidation products at low temperatures, and study the effects of added isoprene (IP) and increased relative humidity (RH). The main focus is on the chemical composition of gas-phase species and (non-size-resolved) ultrafine particles, determined with a nitrate chemical ionization mass spectrometer and a thermal desorption-differential mobility analyzer, and in addition also nucleation rates are reported. Isoprene is observed to affect the chemical composition through an increase in e.g. C5 and C15 compounds, and to suppress new-particle formation as also reported in other studies.

Simultaneous measurements of gas- and particle-phase composition are essential for improving the understanding of biogenic secondary aerosol formation. The manuscript is generally well written and the results are clearly presented. I can recommend the work to be published in ACP after the authors have addressed the following comments:

Specific comments

1. Regarding the discussion on the effects of isoprene on the elemental composition, especially $C_5$ and $C_{15}$ compounds are stated to be increased in intensity. This seems clearer in the case of gas phase, whereas for particle phase the effects seem more diverse and e.g. Figure 3 shows similar increases in the signal intensity for various compounds with carbon content of up to ca. $C_{18}$, $C_{19}$. I cannot clearly distinguish a stronger increase specifically at $C_{15}$ in Figs. 3 or S1; can this be further clarified?

Many thanks for pointing this out. Regarding the isoprene effect, we do observe an increase in the intensity for several species in the particle phase, not only for $C_5$ and $C_{15}$ compounds. We believe that this might be related to the method we apply for evaporating the particles (thermal desorption), possibly fragmentation and chemical reactions are occurring. We are further investigating this phenomenon in a forthcoming study.

We clarified in Section 3.2.1 that $C_5$ and $C_{15}$ compounds correspond to $C_5H_{10}O_{5-7}$ and $C_{15}H_{24}O_{5-10}$ which are observed in both gas and particle phase.

Fig. 3f shows the intensity difference in the particle phase. In contrast with what is observed in the gas phase, the particle phase effects seem more diverse. There is an increase in the intensity difference for several species in the system with isoprene (*αIP-30,20*), such as $C_{4-5}$, $C_{13-16}$ and some $C_{17-19}$ (see Fig. S1 in the supplement). Especially, a distinct group of $C_{15}$ compounds $C_{15}H_{24}O_{5-10}$ and $C_5H_{10}O_{5-7}$ can be identified in the particle and in the gas phase. A previous study has shown that isoprene can enhance particle growth rates despite its negative effect on nucleation (Heinritzi et al., 2020). The identification of $C_{15}$ dimers in nanometer-sized particles in the present study confirms this with a direct measurement. The suppressing effect of isoprene on nucleation will further be discussed in Section 3.4.2. However, isoprene can still contribute to the growth of particles by $C_5$ or by $C_{15}$ compounds. Additionally, these species can be an important fingerprint to identify SOA formation from a mixture of biogenic vapors containing isoprene.

2. It may not be obvious that higher particle growth rates (GR) at larger particle sizes are due to isoprene (Section 3.2.1, last paragraph: "Reaching the same mass with a lower number of particles for the experiment with isoprene (αIP-30,20) compared to α-30,20, means that the growth rates at larger sizes (> 15 nm) are higher in the presence of isoprene").

Particle GRs can generally be higher at lower particle number concentrations, as the amount of available condensable vapor per particle is higher. Can it be concluded that the enhanced growth at larger sizes is specifically related to isoprene, and not to such dynamic effects?

Yes, we think that the enhanced growth at sizes larger than 15 nm for the experiment *αIP-30,20* is related to isoprene. The mentioned dynamic effects can be ruled out in our opinion as the HOM concentrations are not affected by the presence of the particles. This means that the particle number concentration does not influence the degree of supersaturation. Or in other words, there is not depletion of HOM total (as is shown in Figure 1, fourth panel) as there is a continuous replenishment of the gases in the CLOUD chamber.

Additionally, several studies have reported that some oxidation products from isoprene play an important role in SOA formation (different from new particle formation). We mentioned some studies in Section 1.

Isoprene ($C_5H_8$) is the biogenic vapor with the highest global emission rate. Its estimated emissions are between 500 to 600 Tg per year (Guenther et al., 2006; Sindelarova et al., 2014) and there are many studies that indicate the global importance of isoprene in terms of SOA formation (Surratt et al., 2006; Surratt et al., 2007; Surratt et al., 2010; Paulot et al., 2009; Lin et al., 2012; Riva et al., 2016).

3. Effect of RH (Section 3.2.2): the particle mass concentration is observed to increase at elevated RH at otherwise similar conditions. However, Fig. S3 shows that the α-pinene level is somewhat higher for the experiments with higher RH. Can the higher AP level contribute to the increased particle mass?

Effectively the α-pinene level in *α-50,60* is a factor of ~ 2 higher than *α-50,20* in the steady-state (as shown in Table 1). In the same way, the mass concentration also shows an increase by a factor of 2. In this sense, the α-pinene level might influence the number of new particles that are formed and in some proportion the mass. Nevertheless, the very small particles do not contribute significantly to the mass. Additionally, the total HOM concentration and nucleation rates in the gas phase are similar (see Figure 6).

We think that the α-pinene level probably influences the system, but very likely the bigger effect is due to the change in the relative humidity from 20 to 60 %.

Also, the RH range of 60-100% for the high-RH experiments is rather broad. What is the reasoning for lumping together these different RH values, and is it possible that the RH effects vary within this 60-100% range?

The RH for the experiment classified as high RH, was ~ 60 % and did not vary between 60 and 100 % during the particle collection period. In order to clarify this, we included the RH times series in the fifth panel of Figure S3 and we changed the notation accordingly from *α-50,60-100* to *α-50,60* throughout the whole manuscript.

[Figure]

Figure S3. Experimental overview for pure biogenic new particle formation. First panel: particle size distribution for four different experiments: α-pinene + isoprene at -30 °C and 20 % RH (αIP-30,20); α-pinene at -30 °C and 20 % RH (α-30,20); α-pinene at -50 °C and 20 % RH (α-50,20) and α-pinene at -50 °C and 60% RH (α-50,60). The color scale represents the normalized mass concentration in µg m$^{-3}$. Second panel: Particle number concentration in cm$^{-3}$ measured by the PSM with a cut-off diameter of 1.7 nm and CPC with a cut-off diameter of 2.5 nm. Third panel: Mass concentration in µg m$^{-3}$ (obtained by integrating the normalized mass concentration from the SMPS). Fourth panel: Mixing ratio in ppbv for the biogenic precursor gases, isoprene and α-pinene. Fifth panel: Ozone mixing ratio in ppbv and relative humidity in %. The shaded areas refer to the time where the particles were collected using the TD-DMA.

4. Similarly to the RH experiments, it seems that the AP level during the particle formation event and sample collection of the isoprene experiment is not exactly similar to the experiments without IP; it seems to be lower for the AP-IP set-up (Figure 1, third row). Can this affect the AP vs. AP-IP comparison?

It is correct that the α-pinene levels in the experiments with (*αIP-30,20*) and without (*α-30,20*) isoprene are not identical. In fact, since the precursor gases are continuously added into the chamber, reaching identical levels is challenging. Therefore, for the experiments reported in this work, we normalized the mass spectrum, this means that for each experiment, we divided every signal by the total signal, in this sense we are comparing fractions rather than absolute values.

We think that the data analysis treatment together with the fact that all the other parameters (RH, O₃, time collection, temperature) are as similar as reasonably achievable, allows a comparison of the different experiments.

5. Section 3.3: It would be helpful to list the actual fractions of the different VBS bins instead of only stating that the particle-phase species are mainly LVOC, ELVOC and ULVOC (it also seems that ULVOC is only a minor fraction). This could be a table with the bin fractions given for the different experiments.

We agree as well that ULVOC do not represent the highest fraction. We have calculated the volatility bin fractions and reported these values in Table 2.

Table 2. Bin volatility fractions for the different experiments.

| Experiment | T [°C] | RH [%] | ULVOC [%] | ELVOC [%] | LVOC [%] | SVOC [%] | IVOC [%] |
|---|---|---|---|---|---|---|---|
| αIP-30,20 | -30 | 20 | 8.6 | 28.3 | 59.1 | 3.8 | 0.2 |
| α-30,20 | -30 | 20 | 5.9 | 44.1 | 48.1 | 1.9 | 0 |
| α-50,20 | -50 | 20 | 16.5 | 36.4 | 46.1 | 1.0 | 0 |
| α-50,60 | -50 | 60 | 13.8 | 50.6 | 34.8 | 0.8 | 0 |

In Section 3.3, we have mentioned that LVOC and ELVOC represents the major fractions, and ULVOC only a minor fraction.

We classified the volatility bins according to the regimes proposed by Donahue et al. (2012) and Schervish and Donahue (2020), and calculated the corresponding fractions (Table 2). Overall, the particle phase detected compounds correspond mainly to Low Volatility Organic Compounds (LVOC) and Extremely Low Volatility Compounds (ELVOC) by explaining more than 80 % of the signals, while Ultralow Volatility Organic Compounds (ULVOC) represent only a small fraction (between 6 and 17 %). With this parametrization we are able to approximate the saturation mass concentration for the particle phase compounds measured using the TD-DMA in the CLOUD chamber. For this parametrization we assume that the elemental composition is one of the main parameters to take into account.

6. The $O_3$ level of 100 ppbv seems rather high. How does it compare with typical tropospheric values? This is relevant considering the discussion on the effects of mixing ratios on the composition and nucleation of HOM (Section 3.4.1).

The ozone levels along the vertical profile might vary depending on the latitude and longitude. Nevertheless, we can state that in the upper troposphere, approximately between 8 and 12 km, the $O_3$ level of 100 ppbv falls in the upper limit of the observed range (Oltmans et al., 1996; Crutzen et al., 1999; Staehelin, 2003).

However, the ozone and α-pinene mixing ratios used in this study ($O_3$ = 100 ppbv and α-pinene = 1 - 8 ppbv) are considerable higher than the levels reported in Simon et al. (2020), ~ 30 - 50 ppbv $O_3$ and 0.2 - 1 ppbv α-pinene.

In Section 3.4.1, we discuss the possible effects that high levels of precursor gases might have on the composition and nucleation of HOM.

In order to investigate a possible reason for this finding, we have chosen two representative experiments at -10 °C and 80 to 90 % RH with different levels of α-pinene and ozone. Fig. 7 shows the mass defect plots for the gas phase chemical composition of the oxidation products. In one experiment (Fig. 7a) α-pinene and the ozone mixing ratio were between 0.2 to 0.8 ppbv and 40 to 50 ppbv, respectively, while for the second

experiment (Fig. 7b) the mixing ratios were 2 to 3 ppbv and 100 ppbv, respectively. From Fig. 7c it can be concluded that the formation of HOM with low oxygen content is favored when the α-pinene and ozone mixing ratio are higher (relative to the system at low levels of precursor gases). An explanation for this is that the high concentration of $RO_2$ enhances the terminating reactions before the autoxidation can lead to high oxygen content for the products. As the compounds with low oxygen content tend to have higher saturation vapor pressures, they do not contribute efficiently to new particle formation. For this reason, a given total HOM concentration is not unambiguously tied to a new particle formation rate (even at constant temperature). The magnitude of the precursor gas mixing ratio (more specifically the full volatility distribution of the products and not just the simple measure of total HOM) also needs to be taken into account (see Fig.S6 in the supplement). In summary, the lower $J_{1.7nm}$ values compared with previous studies are very likely due to the higher α-pinene and ozone mixing ratios used in the present study. There are several compounds with low oxygen content that contribute to the total HOM concentration in the gas phase while these do not contribute to the formation of new particles.

7. On P5L149-150, it is stated that particle evaporation before analysis should not be substantial; can this be assessed in a quantitative manner? Are there other uncertainty sources such as different charging efficiencies or transmission of the compounds?

Yes, charging and transmission efficiencies are important sources of uncertainties.

Regarding the charging efficiency, the nitrate CI-APi-TOF mass spectrometer uses nitrate reagent ions $(HNO3)_n\ NO_3^-$ with n = 0-2 for detecting HOM. This ionization scheme favors the detection of organic compounds with high O:C ratio (Hyttinen et al., 2017; Ehn et al., 2017), specifically Simon et al. (2020) reported an O:C ratio > 0.6. In this sense, we might underestimate compounds with lower O:C ratio.

On the other hand, the transmission efficiency of the nitrate CI-APi-TOF has been well characterized. Heinritzi et al. (2016) developed a method that makes the HOM quantification more reliable. We applied this method and corrected particle and gas phase mass spectra regarding the mass dependent transmission efficiency.

We have mentioned it briefly in Section 2.3.

Here the nitrate CI-APi-TOF mass spectrometer data for gas and particle phase have been corrected for background signals and the mass-dependent transmission efficiency in the mass classifier (Heinritzi et al., 2016). The data analysis and processing were performed using IGOR Pro 7 (WaveMetrics, Inc., USA), Tofware (Version 3.2, Aerodyne Inc., USA) and MATLAB R2019b (MathWorks, Inc., USA).

I also agree with Referee #1 that an assessment of the relative contributions of the smallest and the larger particles to the particle-phase mass samples would be very useful.

The calculation was done accordingly and the requested information has been added to Table 1.

For calculating the mass fraction of particles collected on the filament during the TD-DMA collection time we used only SMPS size distributions, in a range from 9 nm to 834 nm. We estimated that the mass of particles with sizes smaller than 9 nm is negligible (by comparing to nanoSMPS data in a size range from 2 to 63 nm). Thus, the mass fraction reported in Table 1 takes only into account SMPS measurements. From these data we derived the mass fractions for the collected particles with a diameter smaller than 15 nm (Mass $_{< 15\ nm}$) and for a diameter larger than 15 nm (Mass $_{> 15\ nm}$).

This calculation is an ideal estimate, it does not take into account uncertainties due to charging efficiency, collection efficiency and transmission efficiency in the TD-DMA as reported in detail by Wagner et al. (2018) for the size-selective mode.

Table 1. Summary of the main parameters for four pure biogenic new particle formation experiments.

| Experiment | Isoprene [ppb] | α-pinene [ppb] | Isoprene-to-monoterpene carbon ratio ($R$) | Ozone [ppb] | T [°C] | RH [%] | HOM total* [molec. cm$^{-3}$] | $J_{1.7}$* [cm$^{-3}$ s$^{-1}$] | Growth rate 3.2-8 nm [nm h$^{-1}$] | Growth rate 5-15 nm [nm h$^{-1}$] | Mass $_{<15\,nm}$ /Mass $_{>15\,nm}$** [%] |
|---|---|---|---|---|---|---|---|---|---|---|---|
| αIP-30,20 | 13.71 | 0.95 | 14.4 | 98.58 | -30 | 20 | 1.50e8 | 7.29 | 18.0 | 22.8 | 0.29 / 99.7 |
|  | 31.38 | 5.12 | 6.1 | 101.56 | -30 | 20 | 3.04e8 | 10.10 | NA | 39.0 |  |
| α-30,20 | ~ 0 | 3.35 | NA | 102.10 | -30 | 20 | 2.20e8 | 23.76 | 76.9 | 77.1 | 0.11 / 99.9 |
| α-50,20 | ~ 0 | 3.04 | NA | 100.55 | -50 | 20 | 6.72e7 | 51.24 | 41.1 | 42.0 | 0.26 / 99.7 |
| α-50,60 | ~ 0 | 7.72 | NA | 110.20 | -50 | 60 | 8.00e7 | 79.17 | 63.4 | 78.4 | 0.09 / 99.9 |

* Run-to-run experimental uncertainties of HOMs is ± 20 % and for $J_{1.7}$ is ± 30 %. NA, Not Applicable. **Mass fraction of particles collected on the filament during the TD-DMA collection time, calculation based on SMPS mass distributions.

8. P11L350: The meaning of "GCR conditions" is not explained; please clarify.

The requested information about GCR conditions has been added accordingly in Section 2.1.

In order to promote particle production from ions, Galactic Cosmic Rays conditions (GCR) can be achieved by turning off the high voltage field (30 kV m$^{-1}$). The equilibrium ion-pair concentration in the chamber due to GCR is around 700 cm$^{-3}$ (Kirkby et al., 2016).

The experiments relevant for this work were done under GCR conditions and in a flow-through mode with continuous addition of the reactants, performed at -50 °C and -30 °C, at low and high relative humidity to simulate pure biogenic new particle formation at a range of free tropospheric conditions.

Technical corrections

P6L174: The particle formation rate is said to be defined as the flux of particles of a certain size as a function of time, but presumably the reported rates are not actually time dependent; "as a function of time" should thus be removed, for clarity.

We have removed "as a function of time" as suggested.

The nucleation rate ($J_{dp}$), which is defined as the flux of particles of a certain size, is calculated using the method proposed by Dada et al. (2020), see equation (9) therein.

P6L202: It may be more appropriate to write "this is in line with the results of Kiendler-Scharr et al…." instead of "this confirms the results of …"

We have modified the sentence as suggested.

This is in line with the results of Kiendler-Scharr et al. (2009), who first reported the decrease in particle number of the nucleated particles.

P7L226-227: Please reformulate the expression "the gas and particle of α-pinene" and similar occurrences.

We have modified the sentence.

Figure 3 shows mass defect plots of gas and particle phase and the intensity difference between each phase at -30 °C. Figure 3a and Fig. 3d display the gas and particle composition of *α-30,20*, while the gas and particle composition of *αIP-30,20* are shown in Fig. 3b and Fig. 3e, respectively. As both phases were measured with the same instrument, they can be directly inter-compared.

P8L245: Please change "nSEMS" to "nSMPS" (?).

For determining the growth rates, we used the nano-Scanning Electrical Mobility Spectrometer (nSEMS) as indicated not the nSMPS.

P8L254: Change "growth at higher sizes" to "growth at larger sizes".

We have modified the sentence as suggested.

Most likely, isoprene might enhance growth at larger sizes (> 15 nm) in the present study.

P9L268: The term "mass distribution" (here referring to particle mass size distribution?) may be a bit misleading as it might be confused with elemental composition or volatility distribution; please reformulate.

We have changed "mass distribution" to "particle mass size distribution".

This observation can likely be attributed to a change in the particle mass size distribution (see Fig. S3 in the supplement), which indicates that at similar α-pinene and ozone mixing ratio, and the same temperature, the particle mass concentration increases possibly due to the effect of the relative humidity in the system.

Figure 2 and similar plots: also the particle-phase fractions should preferably be written as positive instead of negative numbers (even if they are presented on the "negative" axis).

For Figures 2, S1 and S2, we have modified the negative axis to positive numbers as suggested.

Caption of Figure 2: For clarity, "each system" could be changed to "each system and phase".

We have modified the sentence as suggested, in the captions of Figure 2, S1 and S2.

Caption of Figure 3 and similar occurrences: the expression "mass defect plots of gas and particle phase and the intensity difference between them" is misleading; this sounds like the intensity difference between the gas and particle phases instead of the difference between the experiments. Please reformulate.

We have changed "mass defect plots of gas and particle phase and the intensity difference between them" to "mass defect plots of gas and particle phase and the intensity difference between each phase", in Figures 3 and 4.

Legend of Figure 6 and similar occurrences: Please change "GRC" to "GCR".

We have attended this comment accordingly.

[Figure]

Figure 6. Pure biogenic nucleation rates of pure α-pinene at 1.7 nm diameter against total HOM concentration at different temperatures. HOM total is defined as the sum of $C_5$, $C_{10}$, $C_{15}$ and $C_{20}$ carbon classes. Triangles represent Galactic Cosmic Rays (GCR) conditions and circles represent Neutral conditions. Data points at -50 °C, -25 °C, -10 °C, +5 °C and +25 °C are from Simon et al. (2020). The points with orange marker on the background are the contribution of this study (experiments α-30,20, α-50,20 and α-50,60). Solid and dashed lines represent power-law fits to GCR and Neutral conditions. Bars indicate 1 σ run-to-run experimental uncertainty. The overall systematic scale uncertainty of HOM of +78 % and -68 % is not shown.

[Figure]

Figure 8. Pure biogenic nucleation rates at 1.7 nm diameter against total HOM concentration at different temperatures for α-pinene and α-pinene + isoprene systems. HOM total is defined as the sum of $C_5$, $C_{10}$, $C_{15}$ and $C_{20}$ carbon classes. Triangles represent Galactic Cosmic Rays (GCR) conditions. Data points at +5 °C and +25 °C are from Kirkby et al. (2016) and Heinritzi et al., (2020). Data points at -25 °C are from Simon et al., (2020). The value of isoprene-to-monoterpene carbon ratio (R) varies from 1.5 to 6.5 for Heinritzi et al., (2020), and R equals to 14.4 and 6.1 for this study. The points with orange marker on the background indicate the contribution of this work. Solid lines represent power-law fits to GCR conditions of the systems with α-pinene only and dashed lines are the power-law fits of the systems with isoprene added. Bars indicate 1 σ run-to-run experimental uncertainty. The overall systematic scale uncertainty of HOM of +78 % and -68 % is not shown.

Caption of Figure 8 and similar occurrences: Please change "galactic comic rays" to "galactic cosmic rays". :-)

We have attended this comment accordingly.

Figure 8. Pure biogenic nucleation rates at 1.7 nm diameter against total HOM concentration at different temperatures for α-pinene and α-pinene + isoprene systems. HOM total is defined as the sum of $C_5$, $C_{10}$, $C_{15}$ and $C_{20}$ carbon classes. Triangles represent Galactic Cosmic Rays (GCR) conditions. Data points at +5 °C and +25 °C are from Kirkby et al. (2016) and Heinritzi et al., (2020). Data points at -25 °C are from Simon et al., (2020). The value of isoprene-to-monoterpene carbon ratio (R) varies from 1.5 to 6.5 for Heinritzi et al., (2020), and R equals to 14.4 and 6.1 for this study. The points with orange marker on the background indicate the contribution of this work. Solid lines represent power-law fits to GCR conditions of the systems with α-pinene only and dashed lines are the power-law fits of the systems with isoprene added. Bars indicate 1 σ run-to-run experimental uncertainty. The overall systematic scale uncertainty of HOM of +78 % and -68 % is not shown.

Caption of Figure S4: Please state that this is Figure 5 in linear scale.

We added this information as suggested

Figure S4. TD-DMA Volatility distribution of the measured oxidation products in the particle phase for four different experiments (Figure 5 in linear scale): (a) α-pinene at -30 °C and 20 % RH (α-30,20); (b) α-pinene + isoprene at -30 °C and 20 % RH (αIP-30,20); (c) α-pinene at -50 °C and 20 % RH (α-50,20) and (d) α-pinene at -50 °C and 60-100 % RH (α-50,60-100). Every individual volatility bin includes the sum of the intensity for the oxidation products normalized by the total signal in each system. Every individual volatility bin is defined at 300 K, shifted, and widened according to their corresponding temperature. The color bands in the background indicate the volatility regimes as in Donahue et al. (2012) and in Schervish and Donahue (2020). The normalized intensity is dimensionless. Nevertheless, it should be noted that the particle phase signal is given in normalized counts per second integrated over the evaporation time [ncps. s].

Figure S5: Why are the orange shades triangle-shaped?

The orange shades indicate the new results reported in this work. Since Figure S5 shows the data points with circle markers, we changed as well to circles for consistency.

[Figure]

Figure S5. Yield of total HOM as a function of temperature for pure α-pinene systems. Data points at -50 °C, -25 °C, -10 °C, +5 °C and +25 °C are from Simon et al. (2020) and the data points with the orange shadows are the contribution of this study (α-30,20, α-50,20 and α-50,60 and complementary pure α-pinene experiments at +5 °C and at -10 °C). The level of precursor gases from Simon et al., (2020) were 0.2 to 0.8 ppbv for α-pinene and 40 to 50 ppbv for ozone, while for experiments reported here (shown with orange shadows), the mixing ratios were 1 to 8 ppbv for α-pinene and 100 ppbv for ozone.

Caption of Figure S6: Please explain the meaning of "overflow bin" and why the values of the first bin are both negative and positive.

The overflow bin contains the volatility bins with a saturation concentration lower than -11 ug m$^{-3}$.

Figure S6. Volatility distribution of gas phase for two different systems and the normalized difference between them. Gas phase is measured with a Nitrate CI-APi-TOF Mass Spectrometer. Every individual volatility bin includes the normalized intensity difference (high-low)/low, which is calculated based on intensity of the system at high level of precursor gases (α-pinene was 2 - 3 ppbv and Ozone ~ 100 ppbv) and on the system at low level of precursor gases (α-pinene was 0.2 - 0.8 ppbv and Ozone ~ 40 - 50 ppbv). Every individual volatility bin is defined at 300 K, shifted, and widened according to their corresponding temperature. The lowest bin is an overflow bin (which contains the volatility bins with saturation concentration < -11 ug m$^{-3}$.). The color bands in the background indicate the volatility regimes as in Donahue et al. (2012) and in Schervish and Donahue (2020).

All the positive and negative bins with saturation values < -11 ug m$^{-3}$ have been summed up and represented in the volatility bin overflow at - 11 ug m$^{-3}$.

[revised manuscript text omitted]

---

## Editor Decision (ED1)

The following corrections are suggested to improve the readability and clarity of the manuscript, primarily in the abstract and introduction. Line numbers refer to manuscript showing tracked changes.

General comments:
Abbreviations such as "NPF" and "SOA" are used inconsistently throughout. It recommended to check that once these have been introduced they are used throughout the manuscript.

It is a little unclear whether the CIMS that is used with the TD-DMA is the same CIMS that is used for the gas phase (CI-APi-TOF). I think part of the confusion is that when discussed with the TD-DMA it is often referred to as the chemical ionization time-of-flight mass spectrometer (e.g., line 141) and not CI-APi-TOF. I think maybe this should just be CI-APi-TOF throughout.

The ratio, R, of isoprene to $\alpha$-pinene is sometimes written as "R" and sometimes as "$R$".

Specific comments:
The outstanding questions or hypotheses that this research seeks to address are a little unclear based on the abstract and the introduction. The sentences in the abstract are a bit disconnected, as are the paragraphs in the introduction. Some specific examples are given below as well as some specific suggestions for revision.

Abstract
Suggested revision of the abstract: "Biogenic organic precursors play an important role in atmospheric new particle formation (NPF). One of the major precursor species is $\alpha$-pinene, which upon oxidation can form a suite of products covering a wide range of volatilities. Highly Oxidized Organic Molecules (HOMs) comprise a fraction of the oxidation products formed. While it is known that HOMs contribute to Secondary Organic Aerosol (SOA) formation, including NPF, they have not been well studied in newly formed particles due to their very low mass concentrations. Here we present gas- and particle-phase chemical composition data from experimental studies of $\alpha$-pinene oxidation, including in the presence of isoprene, at temperatures (-50 °C and -25 °C) and relative humidities (20% and 60%) relevant in the upper free troposphere. The measurements took place at the CERN Cosmics Leaving Outdoor Droplets (CLOUD) chamber. The particle chemical composition was analyzed by a Thermal Desorption-Differential Mobility Analyzer (TD-DMA) coupled to a nitrate chemical ionization time-of-flight mass spectrometer (CIMS). CIMS was used for particle- and gas-phase measurements applying the same ionization and detection scheme."

Line 61: change "are" to "were": "…$C_{15}$ compounds ($C_{15}H_{24}O_{5-10}$) were detected."

No further changes to abstract are suggested.

Introduction
Lines 75-76, suggested revision: "…and are thus relevant for Secondary Organic Aerosol (SOA) formation, including New Particle Formation (NPF), due to gas-particle partitioning.

Lines 80-81 do not make sense as written. The sentence suggests that rapid growth occurs in $\alpha$-pinene ozonolysis experiments across temperatures, but that there is also a reduction in the extent of autoxidation. It isn't really clear what is determined…the rapid growth? A decrease with temperature? An increase with temperature? Maybe it is being suggested that rapid growth is observed across temperatures because of compensating effects-higher autoxidation at higher T and increased partitioning at lower T due to decreased vapor pressures. This is clearer in the discussion of Simon, but is quite confusing in reference to Stolzenburg.

After the first paragraph introducing CCN and HOM, it is suggested that the authors then introduce isoprene and alpha-pinene and give an overview of what has been studied and where the gaps are. One suggestion is: "Isoprene ($C_5H_8$) has the highest global emission rate and many studies have demonstrated the importance of isoprene in terms of SOA formation. $\alpha$-Pinene ($C_{10}H_{16}$), while less abundant, is one of the most commonly observed and prominent contributors to biogenic SOA. SOA formation has been well studied in isoprene and $\alpha$-pinene systems. The role of HOMs in SOA formation and NPF also has been explored in $\alpha$-pinene and $\alpha$-pinene with isoprene systems. However, much less is known about the particle-phase composition of HOMs in these systems and the specific controls particle formation and growth rates, including as a function of temperature and the ratio of isoprene to $\alpha$-pinene. "

A summary of the Kirkby, Stolzenburg, Kiendler-Scharr, etc. studies could then follow. The last paragraph could then be that "In order to better understand  the roles of isoprene and temperature on HOM formation and associated rates of NPF, we present…."

Not sure that the sentence starting on line 106 "Additionally, some studies…" is needed. It does not add much to the specific discussion on HOMs, SOA, NPF, etc.

---

## Author Response (AR2)

*Chemical composition of nanoparticles from α-pinene nucleation and the influence of isoprene and relative humidity at low temperature* by Caudillo et al., Atmos. Chem. Phys. Discuss., https://doi.org/10.5194/acp-2021-512

Dear Kelley Barsanti,

we very much appreciate the editorial comments and suggestions you have made for improving our Manuscript. Please, find below our point-by-point reply to your comments. Our answers are in blue and any reference to the main text in the manuscript can be found in red. Line numbers refer to the manuscript showing tracked changes.

**Editor**
The following corrections are suggested to improve the readability and clarity of the manuscript, primarily in the abstract and introduction. Line numbers refer to manuscript showing tracked changes.

**General comments:**
Abbreviations such as "NPF" and "SOA" are used inconsistently throughout. It recommended to check that once these have been introduced, they are used throughout the manuscript.

We have introduced New Particle Formation (NPF) and Secondary Organic Aerosol (SOA) within the Abstract and Introduction. Later on, NPF and SOA are used as acronyms only.

Lines 48-52
**Abstract.** Biogenic organic precursors play an important role in atmospheric new particle formation (NPF). One of the major precursor species is α-pinene, which upon oxidation can form a suite of products covering a wide range of volatilities. Highly Oxygenated Organic Molecules (HOM) comprise a fraction of the oxidation products formed. While it is known that HOM contribute to Secondary Organic Aerosol (SOA) formation, including NPF, they have not been well studied in newly formed particles due to their very low mass concentrations.

In the Introduction, lines 84-87
These compounds possess low saturation vapor pressures and are thus relevant for Secondary Organic Aerosol (SOA) formation, including New Particle Formation (NPF), due to gas-to-particle partitioning.

It is a little unclear whether the CIMS that is used with the TD-DMA is the same CIMS that is used for the gas phase (CI-APi-TOF). I think part of the confusion is that when discussed with the TD-DMA it is often referred to as the chemical ionization time-of-flight mass spectrometer (e.g., line 141) and not CI-APi-TOF. I think maybe this should just be CI-APi-TOF throughout.

Thank you very much for pointing this out. One of the main goals of our Manuscript is to show that with the same chemical ionization technique we are able to measure the particle and the gas phase. In this regard, it is very important to clarify that the TD-DMA is coupled to the same mass spectrometer for measuring the gas phase. We consider that this can be made clear by adding the acronym (CI-APi-TOF) in line 165.

The particle chemical composition was analyzed by the Thermal Desorption-Differential Mobility Analyzer (TD-DMA) coupled to a nitrate chemical ionization-atmospheric-pressure-interface-time-of-flight (CI-APi-TOF) mass spectrometer.

Besides, we mention it as one of the main conclusions, lines 405-409.

In this study, we demonstrated the suitability of the Thermal Desorption-Differential Mobility Analyzer (TD-DMA) coupled to a nitrate chemical ionization-atmospheric-pressure-interface-time-of-flight (CI-APi-TOF) mass spectrometer for measuring HOM in newly formed nano aerosol particles. Together with the nitrate CI-APi-TOF mass spectrometer, this set up is capable of measuring the gas and particle phase, allowing a direct comparison as both measurements use the identical chemical ionization and detector.

The ratio, R, of isoprene to α-pinene is sometimes written as "R" and sometimes as "*R*".
We have changed R by *R* accordingly.

**Specific comments:**
The outstanding questions or hypotheses that this research seeks to address are a little unclear based on the abstract and the introduction. The sentences in the abstract are a bit disconnected, as are the paragraphs in the introduction. Some specific examples are given below as well as some specific suggestions for revision.

**Abstract**
Suggested revision of the abstract: "Biogenic organic precursors play an important role in atmospheric new particle formation (NPF). One of the major precursor species is α-pinene, which upon oxidation can form a suite of products covering a wide range of volatilities. Highly Oxidized Organic Molecules (HOMs) comprise a fraction of the oxidation products formed. While it is known that HOMs contribute to Secondary Organic Aerosol (SOA) formation, including NPF, they have not been well studied in newly formed particles due to their very low mass concentrations. Here we present gas- and particle-phase chemical composition data from experimental studies of α-pinene oxidation, including in the presence of isoprene, at temperatures (-50 °C and -25 °C) and relative humidities (20% and 60%) relevant in the upper free troposphere. The measurements took place at the CERN Cosmics Leaving Outdoor Droplets (CLOUD) chamber. The particle chemical composition was analyzed by a Thermal Desorption-Differential Mobility Analyzer (TD-DMA) coupled to a nitrate chemical ionization time-of-flight mass spectrometer (CIMS). CIMS was used for particle- and gas-phase measurements applying the same ionization and detection scheme."

We very much prefer the suggested revision of the abstract. We have attended this accordingly and added some minor modifications.

**Abstract.** Biogenic organic precursors play an important role in atmospheric new particle formation (NPF). One of the major precursor species is α-pinene, which upon oxidation can form a suite of products covering a wide range of volatilities. Highly Oxygenated Organic Molecules (HOM) comprise a fraction of the oxidation products formed. While it is known that HOM contribute to Secondary Organic Aerosol (SOA) formation, including NPF, they have not been well studied in newly formed particles due to their very low mass concentrations. Here we present gas and particle phase chemical composition data from experimental studies of α-pinene oxidation, including in the presence of isoprene, at temperatures (-50 °C and -30 °C) and relative humidities (20 % and 60 %) relevant in the upper free troposphere. The measurements took

place at the CERN Cosmics Leaving Outdoor Droplets (CLOUD) chamber. The particle chemical composition was analyzed by a Thermal Desorption-Differential Mobility Analyzer (TD-DMA) coupled to a nitrate chemical ionization-atmospheric-pressure-interface-time-of-flight (CI-APi-TOF) mass spectrometer. CI-APi-TOF was used for particle and gas phase measurements applying the same ionization and detection scheme. Our measurements revealed the presence of $C_{8-10}$ monomers and $C_{18-20}$ dimers as the major compounds in the particles (diameter up to ~ 100 nm). Particularly, for the system with isoprene added, $C_5$ ($C_5H_{10}O_{5-7}$) and $C_{15}$ compounds ($C_{15}H_{24}O_{5-10}$) were detected. This observation is consistent with the previously observed formation of such compounds in the gas phase. However, although the $C_5$ and $C_{15}$ compounds do not easily nucleate, our measurements indicate that they can still contribute to the particle growth at free tropospheric conditions. For the experiments reported here, most likely isoprene oxidation products enhance the growth of particles larger than 15 nm. Additionally, we report on the nucleation rates measured at 1.7 nm ($J_{1.7nm}$) and compared with previous studies, we found lower $J_{1.7nm}$ values, very likely due to the higher α-pinene and ozone mixing ratios used in the present study.

Line 61: change "are" to "were": "…C15 compounds (C15H24O5-10) were detected."
No further changes to abstract are suggested.

We have changed "are" to "were".

Particularly, for the system with isoprene added, $C_5$ ($C_5H_{10}O_{5-7}$) and $C_{15}$ compounds ($C_{15}H_{24}O_{5-10}$) were detected.

**Introduction**
Lines 75-76, suggested revision: "…and are thus relevant for Secondary Organic Aerosol (SOA) formation, including New Particle Formation (NPF), due to gas-particle partitioning.

We have modified the sentence as suggested.

These compounds possess low saturation vapor pressures and are thus relevant for Secondary Organic Aerosol (SOA) formation, including New Particle Formation (NPF), due to gas-to-particle partitioning.

Lines 80-81 do not make sense as written. The sentence suggests that rapid growth occurs in α-pinene ozonolysis experiments across temperatures, but that there is also a reduction in the extent of autoxidation. It isn't really clear what is determined…the rapid growth? A decrease with temperature? An increase with temperature? Maybe it is being suggested that rapid growth is observed across temperatures because of compensating effects-higher autoxidation at higher T and increased partitioning at lower T due to decreased vapor pressures. This is clearer in the discussion of Simon, but is quite confusing in reference to Stolzenburg.

We have modified the paragraph as suggested.

Regarding α-pinene studies, Stolzenburg et al. (2018) reported α-pinene dark ozonolysis experiments at +25 °C, +5 °C, and -25 °C, and showed that the rapid growth of organic particles is observed across these temperatures, higher T leads to a faster autoxidation while lower T leads to an increased partitioning due to decreased vapor pressures. Furthermore, Simon et al. (2020), extended the study of α-pinene gaseous oxidation products to even lower temperatures from +25 °C to -50 °C, showing that the oxygen to carbon

ratio (O:C) and the yield for HOM formation decrease as the temperature decreases, whereas the reduction of volatility compensates this effect by increasing the nucleation rates at lower temperatures.

After the first paragraph introducing CCN and HOM, it is suggested that the authors then introduce isoprene and alpha-pinene and give an overview of what has been studied and where the gaps are. One suggestion is: "Isoprene ($C_5H_8$) has the highest global emission rate and many studies have demonstrated the importance of isoprene in terms of SOA formation. α-Pinene ($C_{10}H_{16}$), while less abundant, is one of the most commonly observed and prominent contributors to biogenic SOA. SOA formation has been well studied in isoprene and α-pinene systems. The role of HOMs in SOA formation and NPF also has been explored in α-pinene and α-pinene with isoprene systems. However, much less is known about the particle-phase composition of HOMs in these systems and the specific controls particle formation and growth rates, including as a function of temperature and the ratio of isoprene to α-pinene. "

A summary of the Kirkby, Stolzenburg, Kiendler-Scharr, etc. studies could then follow. The last paragraph could then be that "In order to better understand the roles of isoprene and temperature on HOM formation and associated rates of NPF, we present…."

We very much value this suggestion; we think that it makes the introduction clearer and have therefore adjusted it as proposed.

Approximately half of the global Cloud Condensation Nuclei (CCN) are produced by nucleation (Merikanto et al., 2009; Gordon et al., 2017). In particular, biogenic emissions of Volatile Organic Compounds (VOCs) play an important role in the formation of aerosol particles. The chemical reactions involving VOCs can lead to the formation of Highly Oxygenated Organic Molecules (HOM), which can be described as a class of organic compounds that are formed under atmospherically relevant conditions by gas phase autoxidation involving peroxy radicals (Ehn et al., 2014; Bianchi et al., 2019). These compounds possess low saturation vapor pressures and are thus relevant for Secondary Organic Aerosol (SOA) formation, including New Particle Formation (NPF), due to gas-to-particle partitioning.

Isoprene ($C_5H_8$) has the highest global emission rate and many studies have demonstrated the importance of isoprene in terms of SOA formation (Surratt et al., 2006; Surratt et al., 2007; Surratt et al., 2010; Paulot et al., 2009; Lin et al., 2012; Riva et al., 2016). α-Pinene ($C_{10}H_{16}$), while less abundant, is one of the most commonly observed and prominent contributors to biogenic SOA due to its ability to form HOM that nucleate on their own under atmospheric conditions (Kirkby et al., 2016; Tröstl et al., 2016). The formation of SOA has been well studied in isoprene and α-pinene systems. The role of HOM in SOA formation and NPF also has been explored in α-pinene and α-pinene with isoprene systems. However, much less is known about the particle phase composition of HOM in these systems and the specific controls particle formation and growth rates, including as a function of temperature and the ratio of isoprene to α-pinene.

Regarding α-pinene studies, Stolzenburg et al. (2018) reported α-pinene dark ozonolysis experiments at +25 °C, +5 °C, and -25 °C, and showed that the rapid growth of organic particles is observed across these temperatures, higher T leads to a faster autoxidation while lower T leads to an increased partitioning due to decreased vapor pressures. Furthermore, Simon et al. (2020), extended the study of α-pinene gaseous oxidation products to even lower temperatures from +25 °C to -50 °C, showing that the oxygen to carbon ratio (O:C) and the yield for HOM formation decrease as the temperature decreases, whereas the reduction of volatility compensates this effect by increasing the nucleation rates at lower temperatures.

Kiendler-Scharr et al. (2009) presented observations at 15 °C of a significant decrease in particle number and volume concentration by the presence of isoprene in an experiment under plant-emitted VOCs conditions. Subsequently, McFiggans et al. (2019) showed that isoprene, carbon monoxide, and methane can each suppress aerosol mass and the yield from monoterpenes in mixtures of atmospheric vapors. Recently, a study by Heinritzi et al. (2020) revealed that the presence of isoprene in the α-pinene system suppresses new particle formation by altering the peroxy-radical termination reactions and inhibiting the formation of those molecules needed for the first steps of cluster and particle formation (species with 19 to 20 carbon atoms).

Despite the difficulties in measuring the nanoparticle chemical composition due to their very small mass, there have been several efforts for designing and improving techniques to face this problem. Some particle phase studies exist that report the chemical composition of newly formed nanoparticles. For instance, Kristensen et al. (2017), measuring at -15 °C and +20 °C, showed an increased contribution of less oxygenated species to α-pinene SOA particles formed from ozonolysis at sub-zero temperatures. Ye et al. (2019) measured the particle phase chemical composition from α-pinene oxidation between -50 °C and +25 °C with the FIGAERO inlet (Lopez-Hilfiker et al., 2014). They found that during new particle formation from α-pinene oxidation, gas phase chemistry directly determines the composition of the condensed phase. Highly Oxygenated Organic Molecules are much more abundant in particles formed at higher temperatures, shifting the compounds towards higher O:C and lower volatilities. Additionally, some studies addressing the chemical composition, volatility, and viscosity of organic molecules have provided important insights into their influence on the climate (Huang et al., 2018; Reid et al., 2018; Champion et al., 2019).

Here, we present the results from gas and particle phase chemical composition measurements for a system where α-pinene was oxidized to simulate pure biogenic new particle formation at free tropospheric conditions in a range from -50 °C to -30 °C. The data are further compared to the mixed system of α-pinene and isoprene in order to better understand the partitioning processes. The particle chemical composition was analyzed by the Thermal Desorption-Differential Mobility Analyzer (TD-DMA) (Wagner et al., 2018), coupled to a nitrate chemical ionization-atmospheric-pressure-interface-time-of-flight (CI-APi-TOF) mass spectrometer. This technique allows a direct comparison between gas and particle phase as both measurements are using the identical chemical ionization source and detector.

Not sure that the sentence starting on line 106 "Additionally, some studies…" is needed. It does not add much to the specific discussion on HOMs, SOA, NPF, etc.

We decided to keep these references, since this information is important for the discussion in Section 3.2.2.

[revised manuscript text omitted]

85  compounds possess low saturation vapor pressures and are thus relevant for Secondary Organic Aerosol (SOA) formation, including New Particle Formation (NPF), and Secondary Organic Aerosol (SOA) formation due to gas-to-particle partitioning. conversion.

Isoprene ($C_5H_8$) has the highest global emission rate and many studies have demonstrated the importance of isoprene in terms of SOA formation (Surratt et al., 2006; Surratt et al., 2007; Surratt et al., 2010; Paulot et al., 2009; Lin et

90  al., 2012; Riva et al., 2016). α-Pinene ($C_{10}H_{16}$), while less abundant, is one of the most commonly observed and prominent contributors to biogenic SOA due to its ability to form HOM that nucleate on their own under atmospheric conditions (Kirkby et al., 2016; Tröstl et al., 2016). The formation of SOA has been well studied in isoprene and α-pinene systems. The role of HOM in SOA formation and NPF also has been explored in α-pinene and α-pinene with isoprene systems. However, much less is known about the particle phase composition of HOM in these systems and the specific controls particle

95  formation and growth rates, including as a function of temperature and the ratio of isoprene to α-pinene.

One of the most prominent biogenic precursors for the formation of particulate material is α-pinene ($C_{10}H_{16}$). It is known that α-pinene oxidation forms HOM that have the ability to nucleate on their own under atmospheric conditions, without the involvement of other trace gases, e. g., sulfuric acid (Kirkby et al., 2016; Tröstl et al., 2016). Regarding α-pinene studies, Stolzenburg et al. (2018) reported showed that the rapid growth of organic particles produced by α-pinene dark

100  ozonolysis experiments at +25 °C, +5 °C, and -25 °C, and showed that the rapid growth of organic particles is observed across these temperatures, higher T leads to a faster autoxidation while lower T leads to an increased partitioning due to decreased vapor pressures 
[revised manuscript text omitted]
.** Biogenic organic precursors play an important role in atmospheric new particle formation (NPF). One of the major precursor species is α-pinene, which upon oxidation can form a suite of products covering a wide range of volatilities. Highly Oxygenated Organic Molecules (HOM) comprise a fraction of the oxidation products formed. While it is known that HOM contribute to Secondary Organic Aerosol (SOA) formation, including NPF, they have not been well studied in newly formed particles due to their very low mass concentrations. Here we present gas and particle phase chemical composition data from experimental studies of α-pinene oxidation, including in the presence of isoprene, at temperatures (-50 ℃ and -30 ℃) and relative humidities (20 % and 60 %) relevant in the upper free troposphere. The measurements took place at the CERN Cosmics Leaving Outdoor Droplets (CLOUD) chamber. The particle chemical composition was analyzed by a Thermal Desorption-Differential Mobility Analyzer (TD-DMA) coupled to a nitrate chemical ionization-atmospheric-pressure-interface-time-of-flight (CI-APi-TOF) mass spectrometer. CI-APi-TOF was used for particle and gas phase measurements applying the same ionization and detection scheme. Our measurements revealed the presence of $C_{8-10}$ monomers and $C_{18-20}$ dimers as the major compounds in the particles (diameter up to ~ 100 nm). Particularly, for the system with isoprene added, $C_5$ ($C_5H_{10}O_{5-7}$) and $C_{15}$ compounds ($C_{15}H_{24}O_{5-10}$) were detected. This observation is consistent with the previously observed formation of such compounds in the gas phase. However, although the $C_5$ and $C_{15}$ compounds do not easily nucleate, our measurements indicate that they can still contribute to the particle growth at free tropospheric conditions. For the experiments reported here, most likely isoprene oxidation products enhance the growth of particles larger than 15 nm. Additionally, we report on the nucleation rates measured at 1.7 nm ($J_{1.7nm}$) and compared with previous studies, we found lower $J_{1.7nm}$ values, very likely due to the higher α-pinene and ozone mixing ratios used in the present study.

**1 Introduction**

Approximately half of the global Cloud Condensation Nuclei (CCN) are produced by nucleation (Merikanto et al., 2009; Gordon et al., 2017). In particular, biogenic emissions of Volatile Organic Compounds (VOCs) play an important role in the formation of aerosol particles. The chemical reactions involving VOCs can lead to the formation of Highly Oxygenated Organic Molecules (HOM), which can be described as a class of organic compounds that are formed under atmospherically relevant conditions by gas phase autoxidation involving peroxy radicals (Ehn et al., 2014; Bianchi et al., 2019). These compounds possess low saturation vapor pressures and are thus relevant for Secondary Organic Aerosol (SOA) formation, including New Particle Formation (NPF), due to gas-to-particle partitioning.

Isoprene ($C_5H_8$) has the highest global emission rate and many studies have demonstrated the importance of isoprene in terms of SOA formation (Surratt et al., 2006; Surratt et al., 2007; Surratt et al., 2010; Paulot et al., 2009; Lin et al., 2012;

Riva et al., 2016). α-Pinene ($C_{10}H_{16}$), while less abundant, is one of the most commonly observed and prominent contributors to biogenic SOA due to its ability to form HOM that nucleate on their own under atmospheric conditions (Kirkby et al., 2016; Tröstl et al., 2016). The formation of SOA has been well studied in isoprene and α-pinene systems. The role of HOM in SOA formation and NPF also has been explored in α-pinene and α-pinene with isoprene systems. However, much less is known about the particle phase composition of HOM in these systems and the specific controls particle formation and growth rates, including as a function of temperature and the ratio of isoprene to α-pinene.

Regarding α-pinene studies, Stolzenburg et al. (2018) reported α-pinene dark ozonolysis experiments at +25 °C, +5 °C, and -25 °C, and showed that the rapid growth of organic particles is observed across these temperatures, higher T leads to a faster autoxidation while lower T leads to an increased partitioning due to decreased vapor pressures. 
[revised manuscript text omitted]